# Text-Aware Image Restoration with Diffusion Models

**Jaewon Min**[*]
KAIST AI

**Jin Hyeon Kim**[*]
KAIST AI

**Paul Hyunbin Cho**
KAIST AI

**Jaeeun Lee**
KAIST AI

**Jihye Park**
Samsung Electronics

**Minkyu Park**
Samsung Electronics

**Sangpil Kim**[†]
Korea University

**Hyunhee Park**[†]
Samsung Electronics

**Seungryong Kim**[†]
KAIST AI

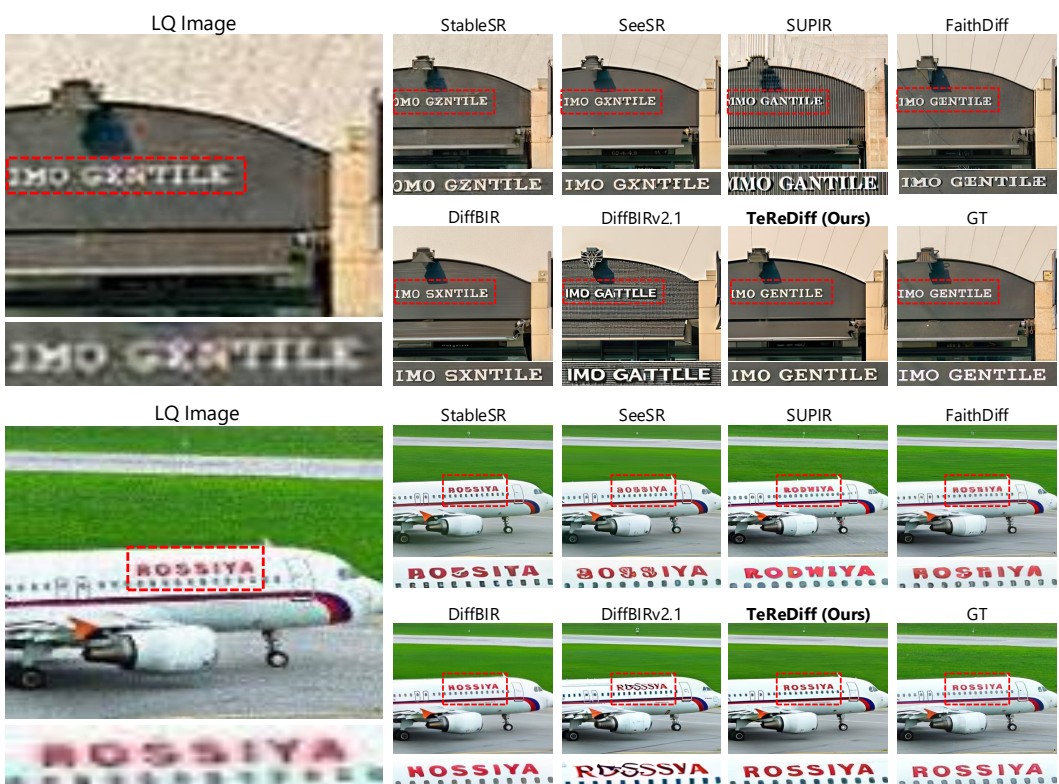

Figure 1: **Text-Aware Image Restoration (TAIR).** Given a low-quality (LQ) image containing degraded texts, our method faithfully restores the original textual content with high legibility and fidelity, whereas previous diffusion-based methods (Blattmann et al., 2023; Wu et al., 2024; Yu et al., 2024; Chen et al., 2024a; Lin et al., 2024) often fail to recover the text regions.

## Abstract

While diffusion models have achieved remarkable success in natural image restoration, they often fail to faithfully recover textual regions, frequently producing plausible yet incorrect text-like patterns, a phenomenon we term *text-image hallucination*. To address this limitation, we propose **Text-Aware Image Restoration (TAIR)**, a task requiring simultaneous recovery of visual content and textual fidelity. For this purpose, we introduce **SA-Text**, a large-scale benchmark of 100K high-quality scene images with dense annotations of diverse and complex text

---

[*]: Equal contribution, [†]: Corresponding authors

instances. We further present a multi-task diffusion model, **TeReDiff**, which leverages internal features of diffusion models to jointly train a text-spotting module with the restoration module. This design allows intermediate text predictions from the text-spotting module to condition the diffusion-based restoration process during denoising, thereby enhancing text recovery. Extensive experiments demonstrate that our approach faithfully restores textual regions, outperforms existing diffusion-based methods, and achieves new state-of-the-art results on TextZoom, an STISR benchmark considered a subtask of TAIR. The code, weights, and dataset will be publicly released.

# 1 INTRODUCTION

Recent studies (Yue et al., 2023; Wang et al., 2024; Lin et al., 2024; Wu et al., 2024; Yu et al., 2024; Chen et al., 2024a) have demonstrated remarkable capabilities in image restoration by leveraging powerful generative priors of diffusion models (Ho et al., 2020; Rombach et al., 2022), achieving superior perceptual quality across various degradation scenarios. However, previous models still struggle to recover text regions, as shown in Fig. 1. Since these models rely on the powerful generative priors of diffusion models, they often synthesize *plausible* text-like images rather than reconstructing the exact characters, leading to *text-image hallucination*. In other words, most image restoration studies have focused on overall perceptual quality without explicitly addressing text readability.

Yet, textual content provides semantic cues that are essential for scenarios such as document digitization (Feng et al., 2023; 2025), street sign understanding (Tabernik & Skočaj, 2019; Ertler et al., 2020), or AR navigation (Li et al., 2020), where even slight distortions can compound into significant information loss. To this end, scene text image super-resolution (STISR) methods tried to improve the perceptual quality and legibility of *cropped* text regions (Dong et al., 2015; Wang et al., 2019b; Xu et al., 2017; Ledig et al., 2017), boosting recognition accuracy, yet its patch-level focus introduces fundamental limitations. The global context is discarded by focusing solely on cropped text regions, thereby ignoring information crucial for overall visual coherence.

To address these limitations, we propose a new task: **Text-Aware Image Restoration (TAIR)**. Unlike STISR methods (Liu et al., 2023a; Noguchi et al., 2024; Singh et al., 2024; Ye et al., 2025), which operate on cropped text images under the single-word assumption, TAIR restores full-scene images while explicitly preserving text fidelity. In contrast to existing image restoration approaches (Wang et al., 2021; Liang et al., 2021; Yue et al., 2023; Wang et al., 2024; Lin et al., 2024; Wu et al., 2024; Yu et al., 2024; Chen et al., 2024a), TAIR necessitates the integration of textual semantics within the restoration process. While concurrent work (Hu et al., 2025) addresses text-aware restoration through segmentation maps, our approach directly leverages linguistic information via text recognition, enabling explicit OCR-guided restoration. However, a primary challenge lies in the absence of suitable datasets.

Existing image restoration benchmarks (Agustsson & Timofte, 2017; Cai et al., 2019; Li et al., 2023; Wei et al., 2020) are not designed for TAIR, making it difficult to train models that align visual restoration with text readability. While some datasets (Singh et al., 2021; Veit et al., 2016; Gupta et al., 2016; Karatzas et al., 2015; Yuliang et al., 2017; Ch'ng & Chan, 2017) provide image-text pairs for text-spotting, they remain suboptimal due to limited scale and quality. These datasets are typically generated synthetically (Gupta et al., 2016) or annotated manually (Singh et al., 2021; Veit et al., 2016; Yuliang et al., 2017; Ch'ng & Chan, 2017), and often suffer from low resolution.

To overcome this, we introduce a large-scale dataset, **SA-Text**, specifically curated for TAIR. The curation pipeline applies text detection and VLM-based recognition, followed by filtering of low-quality samples to yield high-quality crops with accurate annotations. Built on SA-1B (Kirillov et al., 2023), SA-Text comprises 100K images with diverse fonts, sizes, orientations, and complex visual contexts, providing a robust benchmark for evaluating both perceptual restoration quality and text fidelity. To the best of our knowledge, this is the first benchmark addressing both perceptual restoration quality and text fidelity jointly.

Lastly, we propose **Te**xt **Re**storation **Diff**usion model (**TeReDiff**), designed to accomplish TAIR by combining a diffusion-based image restoration model with a text-spotting module. Inspired by recent studies demonstrating the effectiveness of diffusion features in vision downstream tasks (Liu et al., 2023a; Noguchi et al., 2024; Singh et al., 2024; Ye et al., 2025), we directly use diffusion U-Net

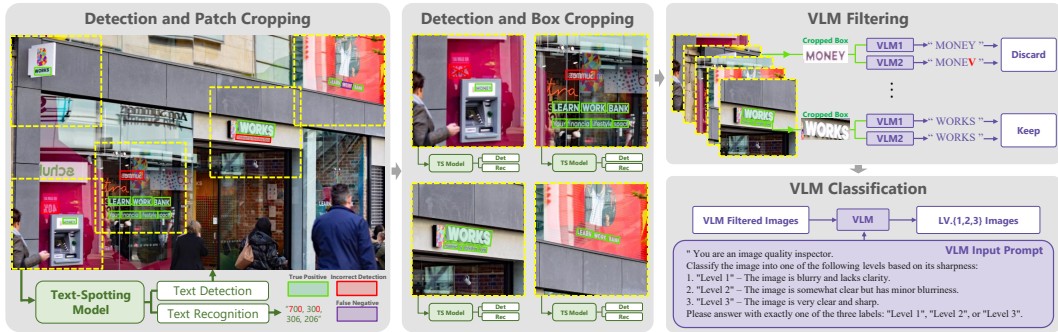

Figure 2: **SA-Text curation pipeline.** A text-spotting module (Huang et al., 2024) detects text regions across the image. To recover missed text, the model is reapplied to smaller patches. Two vision-language models (VLMs) (Bai et al., 2025; Lu et al., 2024) then transcribe the text within the detected boxes, and only patches where both agree are kept. Finally, a single VLM (Bai et al., 2025) classifies each patch by sharpness and blurriness.

features as input to the text-spotting module. This design allows the restoration model to benefit from semantically rich and text-aware representations during training. Furthermore, at inference time, the output of the text-spotting module can be leveraged as input text condition prompts for subsequent denoising steps, thereby jointly enhancing visual quality and text readability.

In summary, our key contributions are as follows:

- We introduce **Text-Aware Image Restoration (TAIR)**, which explicitly requires simultaneous recovery of *scene appearance* and *textual fidelity*.

- We release **SA-Text**, a dataset of 100K high-quality images densely annotated with diverse texts, enabling rigorous evaluation and further research on text-conditioned restoration.

- We propose **TeReDiff**, a multi-task diffusion model that forwards diffusion features to a text-spotting module during training and uses the spotted text as a prompt at inference, enhancing both perceptual quality and legibility.

## 2 RELATED WORK

**Diffusion-based image restoration.**    Advances in diffusion models for generating high-quality images (Ramesh et al., 2021; Rombach et al., 2022; Podell et al., 2023) have extended their use to image restoration (IR) (Lin et al., 2024; Mei et al., 2024; Sahak et al., 2023; Saharia et al., 2022). Unlike GAN-based IR methods (Wang et al., 2021; 2018; Zhang et al., 2021; Che et al., 2016; Mao et al., 2019), which suffer from unstable training and mode collapse, diffusion-based IR offers stable training, robustness, and improved generalization through iterative denoising. SR3 (Saharia et al., 2022) first applied diffusion models to IR, achieving state-of-the-art results on facial and natural image datasets. Subsequent works (Sahak et al., 2023; Wang et al., 2024) further advanced diffusion-based IR by addressing degradation and fidelity challenges.

**Text-spotting.**    Scene text-spotting is the joint task of detecting and recognizing text in natural images. Early methods treated it as two separate tasks: text detection, which localizes regions using region- or segmentation-based techniques (Liao et al., 2017; Liu & Jin, 2017; Wang et al., 2019a; Liao et al., 2020), and text recognition, which interprets the localized content via sequence modeling (Graves et al., 2006; Shi et al., 2016; 2018). Recent detection approaches employ polygon (Ye et al., 2023a) or Bezier curve representations (Liu et al., 2020) to improve localization, while transformer-based architectures inspired by DETR (Carion et al., 2020) have further enhanced detection performance (Ye et al., 2023a; Tang et al., 2022; Zhang et al., 2022). Recognition has concurrently been formulated as an image-to-text translation task, with advances in visual feature extraction and language modeling (Yu et al., 2020; Fang et al., 2021) boosting accuracy.

**Scene text image super resolution.**    Scene text image super-resolution (STISR) aims to enhance the quality and legibility of word-level cropped images. Early approaches employed CNNs (Dong

et al., 2015) and GANs with text-aware losses (Wang et al., 2019b; Xu et al., 2017; Ledig et al., 2017), followed by residual-block architectures (Mou et al., 2020) and high-frequency preservation methods (Wang et al., 2020). More recent transformer-based models incorporated text priors (Ma et al., 2023; 2022), multimodal cues (Zhao et al., 2022), and position- or content-aware losses (Chen et al., 2022). Recently, diffusion-based approaches have also been explored for STISR (Liu et al., 2023a; Noguchi et al., 2024; Singh et al., 2024; Ye et al., 2025).

## 3 SA-Text Dataset

In this section, we present our SA-Text curation pipeline, which comprises a detection stage and a recognition stage, as described in Sec. 3.1. As a fully automated pipeline, it can be easily scaled up to larger datasets with minimal user intervention. We then analyze the resulting dataset, SA-Text, to demonstrate its suitability for TAIR in Sec. 3.2. The accuracy of this automated pipeline was validated through human verification, with additional details provided in Appendix A. Note that although our discussion focuses on English for clarity, the pipeline itself is language-agnostic, with an example on Chinese provided in Appendix A.

### 3.1 Data Curation Pipeline

TAIR requires datasets where text instances are paired with bounding boxes. TextOCR (Singh et al., 2021) provides dense annotations but at low resolution, making it unsuitable for restoration. TextZoom (Wang et al., 2020), built on RealSR (Cai et al., 2019) and SR-RAW (Zhang et al., 2019), offers high-resolution annotations but is limited in scale, with manual construction further hindering scalability. Existing image restoration datasets (Li et al., 2023; Karatzas et al., 2015; Lim et al., 2017) contain high-quality images but lack text annotations, while text-spotting datasets provide dense labels yet mostly low-quality images, leaving both inadequate for TAIR. To overcome this gap, we propose a scalable curation pipeline that produces high-resolution images with dense text annotations, tailored for TAIR. We source images from SA-1B (Kirillov et al., 2023), a corpus of 11M high-resolution images, to ensure quality suitable for restoration.

**Text detection and region cropping.** The first stage of our pipeline applies a dedicated text detection model to high-resolution images to identify text regions. As shown in Fig. 2, small instances are often missed at this resolution, mixed with other incorrect detections. To mitigate these errors, based on the initial detections, we extract $512 \times 512$ crops that fully enclose at least one complete instance and ensure no instance appears in multiple crops. We then re-run the detector on each crop, where the reduced field of view improves recall and refines the incorrect detections, with only minor false positives which are subsequently removed in the VLM-based filtering stage.

**VLM-based text recognition and filtering.** Recent vision-language models (VLMs) (Bai et al., 2025; Lu et al., 2024) show strong text-recognition performance on OCRBench (Liu et al., 2023c), validating their use as recognition backbones. To leverage their accuracy without suffering from localization shortcomings (Ranasinghe et al., 2024), we isolate each text instance via detector polygons and input the cropped patches to two VLMs (Qwen2.5-VL (Bai et al., 2025) and OVIS2 (Lu et al., 2024)). Only instances with identical transcriptions are retained, filtering out misreadings, hard-to-read texts, and false positives. We then apply an additional filtering stage with Qwen2.5-VL (Bai et al., 2025) to remove images with intentional blurring of human faces, license plates, and privacy-sensitive regions, which global blur metrics such as Laplacian filtering often miss. This dual-VLM verification and filtering pipeline enhances the quality of the dataset for TAIR.

| Task | Text-Spotting | | | | | Image Restoration | | | TAIR |
|------|----------|-----------|--------|-----------|---------|-------|------|---------|---------------|
| Dataset | COCOText | ICDAR2015 | CTW1500 | TotalText | TextOCR | LSDIR | DIV2K | Flickr2K | SA-Text (Ours) |
| HQ | ✗ | ✗ | ✗ | ✗ | ✗ | ✓ | ✓ | ✓ | ✓ |
| Text | ✓ | ✓ | ✓ | ✓ | ✓ | ✗ | ✗ | ✗ | ✓ |
| # of Img | 13,880 | 1,000 | 1,000 | 1,255 | 21,778 | 84,991 | 800 | 2,650 | **105,330** |

Table 1: **Comparison with other datasets.**

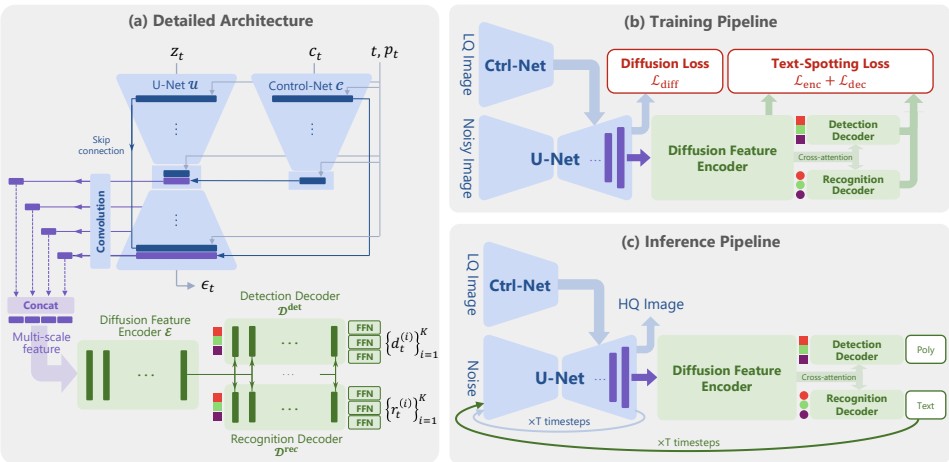

Figure 3: **Overview of the TeReDiff architecture, training, and inference pipeline.** TeReDiff incorporates a text-spotting module into a diffusion-based image restoration framework, using text supervision during training and recognized text as input prompts during inference to enhance text-aware image restoration.

## 3.2 DATASET ANALYSIS

Leveraging our dataset curation pipeline, specifically designed to improve text annotation accuracy, we construct SA-Text 100K from SA-1B (Kirillov et al., 2023). The pipeline is fully automated, making it readily scalable for curating even larger datasets. We compare SA-Text with datasets for text-spotting (Veit et al., 2016; Karatzas et al., 2015; Yuliang et al., 2017; Ch'ng & Chan, 2017; Singh et al., 2021) and image restoration (Li et al., 2023; Karatzas et al., 2015; Lim et al., 2017). As shown in Tab. 1, SA-Text is the only one that provides both high-quality images and explicit text annotations, while also containing the largest number of images among all compared datasets. Since it is curated from the scene-level SA-1B dataset, SA-Text contains diverse text styles and visual contexts.

## 4 TEREDIFF MODEL

The overall architecture of TeReDiff with its training and inference pipelines is illustrated in Fig. 3. A diffusion-based image restoration module enhances the visual and textual fidelity of degraded inputs, while a text-spotting module predicts polygons and transcriptions that are formatted into prompts $p_t$ to guide subsequent denoising steps. These two modules are jointly trained through shared diffusion features, enabling effective interaction and unified optimization of image restoration and text-spotting.

### 4.1 ARCHITECTURE OVERVIEW

**Diffusion-based image restoration module.** Given a low-quality (LQ) image $I_{lq}$, the goal is to recover a high-quality (HQ) image $I_{hq}$ with enhanced visual and textual fidelity. To achieve this, we build on DiffBIR (Lin et al., 2024), comprising a U-Net $\mathcal{U}$ and ControlNet $\mathcal{C}$ (Zhang et al., 2023) as the baseline for the image restoration module. DiffBIR employs a lightweight degradation removal module and a VAE (Kingma et al., 2013) to produce a conditioning latent $c$. Formally, the HQ image $I_{hq}$ is encoded into a latent $z_0$, which undergoes a forward diffusion process that progressively adds noise, yielding a noisy latent $z_t$ at timestep $t$. The conditioning latent $c$ is then concatenated with $z_t$ to form $c_t = \text{concat}(z_t, c)$, which, with text prompts $p_t$, is used to condition the U-Net $\mathcal{U}$ through the ControlNet $\mathcal{C}$.

**Text-spotting module.** Incorporating a text-spotting module within the restoration framework enables the diffusion model to learn text-aware features through gradient supervision. Unlike conventional methods relying on ResNet features (Liu et al., 2020; Yu et al., 2020; Kittenplon et al., 2022; Ye et al., 2023b; Zhang et al., 2022), our module directly exploits semantically rich diffusion features. Specifically, after a single forward pass, we extract intermediate features from the four decoder blocks of U-Net $\mathcal{U}$ (Xu et al., 2023; Tang et al., 2023; Luo et al., 2024). The features from

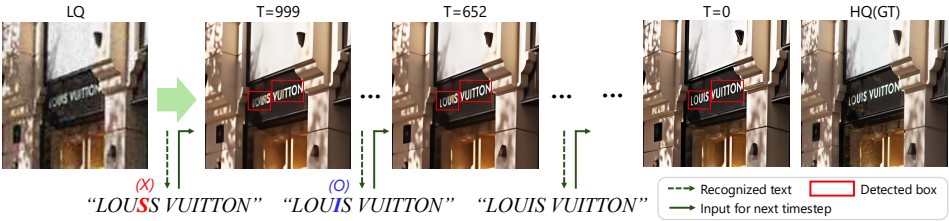

Figure 4: **Textual prompt guidance.** Intermediate text predictions from the text-spotting module is used to condition the restoration module at each denoising step. As shown, the character 'S' is corrected to 'I' during the denoising process, demonstrating that detected boxes and recognized text can be refined as the image is progressively restored.

each decoder block are projected by a convolutional layer to match channel dimensions, and the resulting features are stacked into a multi-scale feature input $F \in \mathbb{R}^{L \times D}$, where $L$ is the number of tokens and $D$ the transformer hidden dimension. This multi-scale input enhances text-spotting while simultaneously benefiting restoration through shared representations. A transformer encoder $\mathcal{E}$ and two decoders, $\mathcal{D}^{\text{det}}$ and $\mathcal{D}^{\text{rec}}$, then decode polygon–character tuples $Y = \{(d_t^{(i)}, r_t^{(i)})\}_{i=1}^{K}$, where $d_t^{(i)}$ are polygon control points, $r_t^{(i)}$ the recognized characters, and $K$ the number of instances above a confidence threshold $T$. Further analysis of leveraging diffusion features in the text-spotting module is provided in Appendix B.

## 4.2 TRAINING

Our training strategy is divided into three sequential stages, each focusing on different components of the framework. In summary, Stage 1 trains the diffusion-based image restoration module, Stage 2 trains the text-spotting module, and Stage 3 jointly optimizes both for mutual benefits. Detailed formulations of the loss functions are provided in Appendix C.

**Stage 1.** In this stage, we focus on training the image restoration module. U-Net $\mathcal{U}$ and ControlNet $\mathcal{C}$ are optimized to learn to remove noise from degraded images, guided by text prompts describing the texts, while the text-spotting module remains frozen. Given the diffusion timestep $t$, prompt $p_t$, and control input $c_t$, the model learns a noise prediction network $\epsilon_\theta$ that estimates the noise added to the HQ latent $z_t$ with diffusion loss.

**Stage 2.** During this stage, we train only the text-spotting module using the conventional loss formulation adopted in transformer-based text-spotting methods (Huang et al., 2024; Zhang et al., 2022; Qiao et al., 2024). The module outputs polygonal regions and character sequences, with bipartite matching (Carion et al., 2020) applied to associate predictions with ground truth. The encoder branch is optimized with classification and localization objectives, while the decoder branches refine polygon boundaries and character transcriptions. This stage preserves the standard training paradigm for text spotting while adapting it to diffusion features. The detailed optimization objectives of the text-spotting module are provided in Appendix C.

**Stage 3.** In the final stage, both the diffusion-based restoration module and the text-spotting module are optimized jointly. This unified training allows restoration to benefit from text-aware supervision and, simultaneously, improves text-spotting performance through cleaner image features. The overall objective combines the restoration loss with text detection and recognition losses, balanced by a weighting factor to encourage mutual gains across tasks.

## 4.3 INFERENCE

**Textual prompt guidance.** At each denoising timestep, the text-spotting module produces text detection and recognition outputs in the form of polygon-character tuples. To further guide the restoration of accurate textual content in the LQ image, the intermediate predicted characters are employed as text conditions for the subsequent timestep. Specifically, at timestep $t + 1$, the characters

| Deg. Level | Model | ABCNet v2 (Liu et al., 2021) | | | | | TESTR (Zhang et al., 2022) | | | | |
| | | Detection | | | End-to-End | | Detection | | | End-to-End | |
| | | Precision(↑) | Recall(↑) | F1-Score(↑) | None(↑) | Full(↑) | Precision(↑) | Recall(↑) | F1-Score(↑) | None(↑) | Full(↑) |
| - | HQ (GT) | 92.16 | 86.85 | 89.43 | 71.79 | 82.05 | 92.59 | 87.81 | 90.13 | 75.90 | 84.18 |
| Level1 | LQ (Lv1) | 89.79 | 29.51 | 44.42 | 24.29 | 34.25 | 84.01 | 30.24 | 44.47 | 25.93 | 34.73 |
| | Real-ESRGAN (Wang et al., 2021) | 83.79 | 43.34 | 57.13 | 21.45 | 30.12 | 85.19 | 41.98 | 56.24 | 22.90 | 31.52 |
| | SwinIR (Liang et al., 2021) | 84.95 | 40.93 | 55.25 | 22.70 | 31.26 | 87.93 | 39.62 | 54.63 | 25.50 | 33.75 |
| | ResShift (Yue et al., 2023) | 81.93 | 40.07 | 53.82 | 20.40 | 28.74 | 88.47 | 35.81 | 50.98 | 22.39 | 30.14 |
| | StableSR (Wang et al., 2024) | 77.90 | 55.44 | 64.78 | 21.93 | 29.29 | 84.44 | 50.68 | 63.34 | 24.02 | 31.84 |
| | DiffBIR (Lin et al., 2024) | 76.29 | 56.44 | 64.88 | 23.14 | 32.73 | 84.00 | 52.13 | 64.34 | 25.51 | 35.47 |
| | DiffBIR† (Lin et al., 2024) | 53.01 | 51.86 | 52.43 | 15.26 | 20.71 | 60.53 | 51.99 | 55.94 | 16.78 | 22.82 |
| | SeeSR (Wu et al., 2024) | 70.00 | 61.88 | 65.69 | 20.16 | 28.63 | 78.85 | 55.76 | 65.32 | 23.31 | 32.82 |
| | SUPIR (Yu et al., 2024) | 43.64 | 49.46 | 46.37 | 14.58 | 19.34 | 53.02 | 46.19 | 49.37 | 17.44 | 22.29 |
| | FaithDiff (Chen et al., 2024a) | 69.16 | 61.51 | 65.12 | 20.44 | 27.78 | 78.80 | 57.12 | 66.23 | 22.50 | 31.59 |
| | TeReDiff (Ours) | 85.29 | 58.34 | 69.29 | 26.59 | 35.69 | 87.50 | 54.90 | 67.47 | 28.19 | 36.99 |
| Level2 | LQ (Lv2) | 87.67 | 22.89 | 36.30 | 20.49 | 27.82 | 78.45 | 23.93 | 36.68 | 20.49 | 27.37 |
| | Real-ESRGAN (Wang et al., 2021) | 81.42 | 41.12 | 54.64 | 18.31 | 24.88 | 84.92 | 38.80 | 53.27 | 19.29 | 27.50 |
| | SwinIR (Liang et al., 2021) | 80.14 | 37.31 | 50.91 | 17.82 | 24.93 | 85.43 | 34.81 | 49.47 | 19.07 | 26.99 |
| | ResShift (Yue et al., 2023) | 81.11 | 35.22 | 49.12 | 17.89 | 25.54 | 85.18 | 32.05 | 46.57 | 17.26 | 26.09 |
| | StableSR (Wang et al., 2024) | 75.49 | 51.95 | 61.55 | 19.55 | 26.69 | 79.03 | 48.87 | 60.39 | 20.06 | 27.68 |
| | DiffBIR (Lin et al., 2024) | 72.96 | 53.94 | 62.03 | 19.60 | 27.52 | 79.69 | 50.50 | 61.82 | 21.64 | 30.36 |
| | DiffBIR† (Lin et al., 2024) | 53.28 | 51.18 | 52.21 | 14.75 | 19.61 | 58.85 | 50.18 | 54.17 | 16.15 | 21.43 |
| | SeeSR (Wu et al., 2024) | 68.93 | 60.65 | 64.53 | 19.48 | 26.72 | 77.50 | 54.49 | 63.99 | 21.83 | 29.17 |
| | SUPIR (Yu et al., 2024) | 42.01 | 45.65 | 43.75 | 13.21 | 17.42 | 53.54 | 43.25 | 47.84 | 15.50 | 19.96 |
| | FaithDiff (Chen et al., 2024a) | 66.62 | 59.34 | 62.77 | 18.94 | 25.99 | 76.16 | 54.17 | 63.31 | 20.98 | 28.56 |
| | TeReDiff (Ours) | 83.02 | 56.30 | 67.10 | 24.42 | 33.23 | 86.95 | 52.86 | 65.75 | 26.39 | 35.13 |
| Level3 | LQ (Lv3) | 85.38 | 13.24 | 22.92 | 12.17 | 16.95 | 76.01 | 15.37 | 25.57 | 12.52 | 17.72 |
| | Real-ESRGAN (Wang et al., 2021) | 72.48 | 28.65 | 41.07 | 11.89 | 16.37 | 76.51 | 27.02 | 39.93 | 12.13 | 18.22 |
| | SwinIR (Liang et al., 2021) | 74.41 | 25.57 | 38.06 | 11.27 | 16.33 | 78.38 | 24.16 | 36.94 | 11.85 | 17.12 |
| | ResShift (Yue et al., 2023) | 75.00 | 22.57 | 34.70 | 10.80 | 15.19 | 81.10 | 20.04 | 32.13 | 9.89 | 15.63 |
| | StableSR (Wang et al., 2024) | 67.63 | 38.08 | 48.72 | 13.34 | 18.91 | 72.21 | 35.22 | 47.35 | 13.65 | 19.68 |
| | DiffBIR (Lin et al., 2024) | 59.30 | 42.20 | 49.31 | 13.88 | 19.39 | 72.27 | 38.98 | 50.65 | 15.61 | 22.67 |
| | DiffBIR† (Lin et al., 2024) | 44.16 | 40.62 | 42.31 | 10.15 | 14.45 | 49.73 | 41.25 | 45.09 | 10.75 | 15.41 |
| | SeeSR (Wu et al., 2024) | 55.06 | 46.83 | 50.61 | 13.38 | 18.47 | 64.95 | 43.93 | 52.41 | 14.93 | 20.88 |
| | SUPIR (Yu et al., 2024) | 31.05 | 34.72 | 32.78 | 9.07 | 11.77 | 40.78 | 32.77 | 36.34 | 11.21 | 14.02 |
| | FaithDiff (Chen et al., 2024a) | 56.04 | 47.91 | 51.66 | 13.69 | 19.01 | 69.44 | 45.01 | 54.62 | 15.40 | 21.18 |
| | TeReDiff (Ours) | 81.76 | 44.11 | 57.30 | 19.61 | 27.50 | 84.50 | 42.02 | 56.13 | 19.92 | 28.34 |

Table 2: **Quantitative results of TAIR on SA-Text.** Each block shows the performance of various image restoration methods under different degradation strengths, evaluated using two text-spotting models. 'None' refers to recognition without a lexicon, and 'Full' denotes recognition with a full lexicon. Best results are in **bold** and second-best are underlined. DiffBIR (Lin et al., 2024)† (denoted as v2.1 in Fig. 1) uses GitHub-released weights that are further trained from those in the original paper.

predicted by the recognition branch of the text-spotting module at the previous timestep $t$, $\{r_t^{(i)}\}_{i=1}^K$, are used as the text prompt $p_{t+1}$. An example of textual prompt guidance is illustrated in Fig. 4.

## 5 EXPERIMENTS

### 5.1 EXPERIMENTAL SETTINGS

**Training and evaluation dataset.** We train our TeReDiff on SA-Text, which contains 100K high-quality 512×512 images. Synthetic degradations are applied using the Real-ESRGAN pipeline (Wang et al., 2021), a method widely adopted in image restoration studies (Yue et al., 2023; Wang et al., 2024; Lin et al., 2024; Wu et al., 2024; Yu et al., 2024). For testing, we curate a separate 1K subset of SA-Text. Similar to DASR (Liang et al., 2023) and FaithDiff (Chen et al., 2024a), we apply three progressively challenging degradation levels to our test set, to evaluate the model in various difficulty settings. We further construct Real-Text, a set of real-world HR-LR pairs extracted from RealSR (Cai et al., 2019) and DRealSR(Wei et al., 2020) with text annotations using our dataset curation pipeline. Lastly, we evaluate TeReDiff on TextZoom (Wang et al., 2020), an STISR benchmark consisting of paired low-resolution (LR) and high-resolution (HR) text images sourced from real-world images.

**Evaluation metrics.** We evaluate text restoration using pretrained text-spotting models (Liu et al., 2021; Zhang et al., 2022), reporting Precision, Recall, and F1-score for detection and F1-score for end-to-end recognition. The evaluation operates in two stages: (1) text detection is assessed using Precision, Recall, and F1-score to measure how well text regions are localized, and (2) for detected text regions, End-to-End metrics evaluate recognition accuracy by comparing transcribed text against ground-truth annotations using exact string matching. This process directly measures linguistic correctness. If restoration produces inaccurate or garbled text, End-to-End F1-scores will be lower. These metrics collectively measure functional legibility, where restoration is successful when words are more readily detected and recognized. This evaluation protocol follows standard practice in text-spotting benchmarks. Image restoration is assessed with reference metrics, using PSNR and

| Model | ABCNet v2 (Liu et al., 2021) | | | | | TESTR (Zhang et al., 2022) | | | | |
|---|---|---|---|---|---|---|---|---|---|---|
| | Detection | | | End-to-End | | Detection | | | End-to-End | |
| | Precision(↑) | Recall(↑) | F1-Score(↑) | None(↑) | Full(↑) | Precision(↑) | Recall(↑) | F1-Score(↑) | None(↑) | Full(↑) |
| HQ (GT) | 90.03 | 85.52 | 87.72 | 72.06 | 79.48 | 90.29 | 85.77 | 87.97 | 74.50 | 81.72 |
| LQ | 89.10 | 44.97 | 59.77 | 42.64 | 50.21 | 85.33 | 51.61 | 64.32 | 47.08 | 55.11 |
| Real-ESRGAN (Wang et al., 2021) | 79.15 | 52.70 | 63.27 | 35.30 | 39.88 | 82.67 | 53.94 | 65.29 | 38.16 | 42.36 |
| SwinIR (Liang et al., 2021) | 80.29 | 47.45 | 59.64 | 38.39 | 42.63 | 82.92 | 47.89 | 60.72 | 39.97 | 44.56 |
| ResShift (Yue et al., 2023) | 81.17 | 33.76 | 47.69 | 30.95 | 34.87 | 82.23 | 39.91 | 53.74 | 35.31 | 39.99 |
| StableSR (Wang et al., 2024) | 79.79 | 59.89 | 68.42 | 41.23 | 47.64 | 82.19 | 60.39 | 69.62 | 42.53 | 49.39 |
| DiffBIR (Lin et al., 2024) | 66.04 | 59.69 | 62.71 | 33.75 | 40.05 | 76.33 | 61.87 | 68.35 | 39.27 | 46.11 |
| DiffBIR† (Lin et al., 2024) | 55.31 | 56.02 | 55.67 | 26.85 | 31.23 | 58.99 | 60.19 | 59.58 | 31.41 | 35.98 |
| SeeSR (Wu et al., 2024) | 68.12 | 63.46 | 65.71 | 37.11 | 43.43 | 74.29 | 62.47 | 67.87 | 40.34 | 46.54 |
| SUPIR (Yu et al., 2024) | 44.00 | 40.56 | 42.21 | 22.29 | 25.03 | 53.08 | 44.47 | 48.39 | 27.25 | 30.59 |
| FaithDiff (Chen et al., 2024a) | 71.21 | 64.50 | 67.69 | 38.81 | 44.28 | 76.90 | 65.20 | 70.57 | 41.64 | 47.97 |
| **TeReDiff (Ours)** | **83.95** | **67.58** | **74.88** | **48.39** | **55.01** | **84.30** | **67.37** | **74.89** | **49.39** | **56.45** |

Table 3: **Quantitative results of TAIR on Real-Text.** Using text-spotting models, we evaluate the accuracy of text detection and recognition of the restored images on Real-Text.

| Dataset | Model | PSNR(↑) | SSIM(↑) | LPIPS(↓) | DISTS(↓) | FID(↓) | NIQE(↓) | MANIQA(↑) | MUSIQ(↑) | CLIPIQA(↑) |
|---|---|---|---|---|---|---|---|---|---|---|
| SA-Text$_{test}$ | DiffBIR | 19.58 | 0.4965 | 0.3636 | 0.2080 | 45.10 | **5.107** | 0.6771 | **73.33** | **0.6589** |
| | DiffBIR† | 16.81 | 0.4638 | 0.4001 | 0.2219 | 47.28 | 5.449 | **0.6890** | 72.55 | 0.6345 |
| | **TeReDiff (Ours)** | **19.71** | **0.5717** | **0.2828** | **0.1702** | **36.94** | 5.452 | 0.6471 | 72.07 | 0.6145 |
| Real-Text | DiffBIR | 23.00 | 0.6516 | 0.4108 | 0.2925 | 87.46 | **7.054** | **0.6147** | **66.84** | **0.5679** |
| | DiffBIR† | 19.11 | 0.5107 | 0.5233 | 0.3127 | 80.59 | 8.42 | 0.5873 | 61.46 | 0.4961 |
| | **TeReDiff (Ours)** | **23.37** | **0.7849** | **0.2848** | **0.2386** | **68.94** | 7.643 | 0.5637 | 62.02 | 0.4545 |

Table 4: **Quantitative results of image restoration.** We evaluate the image quality of our TeReDiff compared with its baseline. The degradation pipeline (Wang et al., 2021) used in prior works is applied. DiffBIR (Lin et al., 2024)† (denoted as v2.1 in Fig. 1) uses GitHub-released weights that are further trained from those in the original paper.

SSIM (Wang et al., 2004) for fidelity, LPIPS (Zhang et al., 2018) and DISTS (Ding et al., 2020) for perceptual quality, and FID (Heusel et al., 2017) for distributional similarity, as well as non-reference metrics including NIQE (Zhang et al., 2015), MANIQA (Yang et al., 2022), MUSIQ (Ke et al., 2021), and CLIPIQA (Wang et al., 2023).

**Implementation details.** The image restoration module is initialized from DiffBIR (Lin et al., 2024), and the text-spotting module from TESTR (Zhang et al., 2022). Training is performed using the AdamW optimizer with default parameters. The learning rate is set to $1 \times 10^{-4}$ for Stage 1 and Stage 2, and $1 \times 10^{-5}$ for Stage 3. Publicly available checkpoints are used for all models. The input LR and output HR images are both $512 \times 512$ in size. Further details can be found in Appendix D.

## 5.2 MAIN RESULTS

**Quantitative comparisons.** Tab. 2 shows detection and recognition metrics on SA-Text. TeReDiff achieves the best F1-score at every level on both text-spotting models. Prior models often lose recognition accuracy at Level2 and Level3, dropping below the raw low-resolution inputs because stronger degradations intensify *text-image hallucination*. In contrast, TAIR consistently restores textual regions, preserving recognition performance even under the strongest degradations. Furthermore, Tab. 3 illustrates the performance of TeReDiff in real-world scenarios. For image quality, Tab. 4 demonstrates that our model outperforms the baseline on reference-based metrics while achieving comparable performance on non-reference metrics. These results confirm that incorporating text-focused restoration objectives does not compromise overall image quality. Although existing image quality metrics are not well-suited to capturing the *text-image hallucination* problems that we aim to address, our text-aware approach maintains competitive image quality while significantly improving text legibility, validating the primary focus of this paper. Additional quantitative comparisons of image quality with other methods are provided in Appendix E.

**Qualitative comparisons.** Fig. 1 shows representative results on SA-Text test set. Previous restoration methods often produce blurry characters and inconsistent stroke widths, which are text-image hallucinations under severe degradations due to reliance on the generative priors of diffusion models. In contrast, TeReDiff consistently restores readable text in challenging regions. These qualitative improvements align closely with the quantitative results in Tab. 3, highlighting our model's

| Method | CRNN (Shi et al., 2016) | | | | MORAN (Luo et al., 2019) | | | | ASTER (Shi et al., 2018) | | | |
|---|---|---|---|---|---|---|---|---|---|---|---|---|
| | Easy | Medium | Hard | Avg. | Easy | Medium | Hard | Avg. | Easy | Medium | Hard | Avg. |
| LR (Bicubic) | 36.4% | 21.1% | 21.1% | 26.8% | 60.6% | 37.9% | 30.8% | 44.1% | 64.7% | 42.4% | 31.2% | 47.2% |
| HR | 76.4% | 75.1% | 64.6% | 72.4% | 91.2% | 85.3% | 74.2% | 84.1% | 94.2% | 87.7% | 76.2% | 86.6% |
| TextDiff (Liu et al., 2023a) | 64.8% | 55.4% | 39.9% | 54.2% | 77.7% | 62.5% | 44.6% | 62.7% | 80.8% | 66.5% | 48.7% | 66.4% |
| TCDM (Noguchi et al., 2024) | 67.3% | 57.3% | 42.7% | 55.7% | 77.6% | 62.9% | 45.9% | 62.2% | 81.3% | 65.1% | 50.1% | 65.5% |
| DCDM (Singh et al., 2024) | 65.7% | 57.3% | 41.4% | 55.5% | 78.4% | 63.5% | 45.3% | 63.4% | 81.8% | 65.1% | 47.4% | 65.8% |
| TextSR (Ye et al., 2025) | 69.9% | 60.0% | 43.9% | 58.7% | 78.7% | 63.2% | 47.1% | 64.0% | 81.0% | 64.6% | 48.7% | 65.8% |
| **TeReDiff (Ours)** | **75.7%** | **75.6%** | **65.8%** | **72.4%** | **92.2%** | **86.0%** | **76.8%** | **85.0%** | **89.6%** | **84.0%** | **73.0%** | **82.2%** |

Table 5: **Quantitative results on TextZoom dataset.** We evaluate performance using standard text recognition models (Shi et al., 2016; Luo et al., 2019; Shi et al., 2018) under the common STISR scheme, noting that TextSR (Ye et al., 2025) is concurrent work.

| Model | Text Condition | TESTR (Zhang et al., 2022) | | | | |
|---|---|---|---|---|---|---|
| | | Detection | | | End-to-End | |
| | | Precision(↑) | Recall(↑) | F1-Score(↑) | None(↑) | Full(↑) |
| Stage1 | Null | 81.77 | 47.37 | 59.99 | 21.24 | 29.79 |
| | Captioner (llava) | 82.01 | 49.82 | 61.99 | 24.76 | 31.70 |
| | Ground-truth | 85.09 | 61.56 | 71.44 | 32.51 | 42.71 |
| Stage3 | Null | 84.47 | 56.21 | 67.50 | 23.46 | 32.72 |
| | Captioner (Ours) | 86.95 | 52.86 | 65.75 | 26.39 | 35.13 |
| | Ground-truth | 86.18 | 61.60 | 71.85 | 33.31 | 43.40 |

(a) **Multi-stage training and text conditioning.** Stage-wise training and text prompting of the restoration module improve text restoration.

| Text Condition | Prompt Style | TESTR (Zhang et al., 2022) | | | | |
|---|---|---|---|---|---|---|
| | | Detection | | | End-to-End | |
| | | Precision(↑) | Recall(↑) | F1-Score(↑) | None(↑) | Full(↑) |
| Captioner (Ours) | Tag | 83.25 | 59.25 | 69.23 | 24.26 | 31.94 |
| | Descriptive(Ours) | 86.95 | 52.86 | 65.75 | 26.39 | 35.13 |
| Ground-truth | Tag | 84.40 | 62.56 | 71.86 | 32.02 | 42.12 |
| | Descriptive(Ours) | 86.18 | 61.60 | 71.85 | 33.31 | 43.40 |

(b) **Prompt style.** The descriptive-style prompting shows better performance than the tag-style prompting.

Table 6: **Ablation study on SA-Text (Level2 degradation).** (a) Extending training to Stage 3 enables the model to learn text-aware features, thereby improving text restoration performance. Moreover, text conditioning using captioner prompts yields further gains compared to null prompts. (b) The prompt style employed in TeReDiff impacts text restoration performance. In particular, descriptive prompts result in greater improvements than tag-style prompts.

effectiveness in enhancing text clarity without compromising overall image quality. Additional qualitative results are provided in Appendix F.

**Comparison to STISR methods.** As shown in Tab. 5, our method establishes a new SOTA on existing STISR methods (Liu et al., 2023a; Noguchi et al., 2024; Singh et al., 2024; Ye et al., 2025), highlighting the effectiveness of training on full-scene images with diverse context. Importantly, our task formulation, TAIR, is hierarchically broader and more challenging than STISR: while STISR assumes small inputs (e.g., $64 \times 16$) with a single word, TAIR processes full-scene images (e.g., $512 \times 512$) containing multiple texts with diverse styles and complex backgrounds. Since STISR benchmarks rely on recognition models for cropped word images, evaluation is restricted to a narrow sub-case of our task. Nevertheless, our strong results under this constrained protocol demonstrate the generality and robustness of our framework, clarifying its distinction from existing STISR methods. Qualitative results on TextZoom (Wang et al., 2020) of TeReDiff is provided in Appendix F.

## 5.3 ABLATION STUDY

We conduct ablation studies on SA-Text test set applied with Level2 degradation to evaluate the effectiveness of multi-stage training and the importance of text prompting. Additional ablations on all levels of SA-Text test set and on Real-Text are provided in Appendix E.

**Multi-stage training.** To assess the effectiveness of multi-stage training for text-aware restoration, we compare the image restoration module trained up to Stage 1 and Stage 3. Advancing training to Stage 3 consistently improves text restoration performance across all text conditions (Null, Captioner, and Ground-truth), as the image restoration module has learned text-aware features by receiving gradients from the text-spotting module, as shown in Tab. 6a. The Null, Captioner, and Ground-Truth settings correspond, respectively, to conditioning the restoration module with no prompt, a text prompt obtained from a captioner (LLaVA (Liu et al., 2023b) or our text-spotting module), and ground-truth text.

**Text prompting.** As shown in Tab. 6a, conditioning on textual content extracted from the LQ image consistently improves restoration quality in both Stage 1 and Stage 3, highlighting the effectiveness of text prompting for TAIR. In this setting, Stage 1 leverages an external captioning module (Liu et al.,

2023b) due to the absence of a trained text-spotting module, whereas Stage 3 employs our trained text-spotting module. Complementary results in Tab. 6b present the text restoration performance of TeReDiff under two different prompt styles. Descriptive prompts yield greater improvements than tag-style prompts when conditioning the diffusion-based restoration module with either predicted or ground-truth texts. For example, given the predicted texts "text1," "text2," and "text3," the descriptive-style adopts the format "A realistic scene where the texts text1, text2, . . . appear clearly on signs, boards, buildings, or other objects," whereas the tag-style uses the simpler format "text1, text2, . . . ." The same trend is observed when ground-truth texts are used as prompts.

## 6 CONCLUSION

We revisit image restoration with a new focus: Text-Aware Image Restoration (TAIR), which targets the recovery of textual content in degraded images. This area remains largely unexplored due to the absence of large-scale, annotated datasets. To address this gap, we introduce SA-Text, a curated dataset that uses VLMs to provide automated supervision for text restoration. By leveraging diffusion features in a joint optimization framework, TeReDiff achieves substantial improvements over existing methods and establishes a new state-of-the-art performance on the STISR task. We hope that TAIR and SA-Text will inspire further research combining text understanding and image restoration.

ACKNOWLEDGMENTS

This research was supported by Institute of Information & communications Technology Planning & Evaluation (IITP) grant funded by the Korea government (MSIT) (RS-2019-II190075, RS-2024-00509279, RS-2025-II212068, RS-2023-00227592, RS-2025-02214479, RS-2024-00457882, RS-2025-25441838, RS-2025-25441838, RS-2025-02214479, RS-2025-02217259) and the Culture, Sports, and Tourism R&D Program through the Korea Creative Content Agency grant funded by the Ministry of Culture, Sports and Tourism (RS-2024-00345025, RS-2024-00333068, RS-2023-00222280, RS-2023-00266509), and National Research Foundation of Korea (RS-2024-00346597).

## REPRODUCIBILITY STATEMENT

Training details are provided in Sec 5.1 and Appendix D. We will release the proposed dataset, the codes, and the model weights to ensure reproducibility.

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

# APPENDIX

This appendix is organized as follows. Sec. A details the SA-Text curation pipeline, with emphasis on handling false negatives and incorrect detections. Sec. B demonstrates the benefits of leveraging diffusion features for training the text-spotting module. Further descriptions of the loss functions are given in Sec. C. Sec. D outlines the implementation of the TAIR framework, including evaluation metrics, model architecture, and training details. Sec. E reports extended quantitative results, including comparisons with baselines across three levels of degradation on SA-Text and on Real-Text, which captures real-world degradation scenarios. We also present the results of a user study assessing human preference for restoration quality. Finally, Sec. F provides qualitative comparisons of TAIR against conventional image restoration and STISR methods, and Sec. G provides additional discussions on TeReDiff. Lastly, Sec. H describes the use of LLMs.

## A    SA-TEXT CURATION PIPELINE

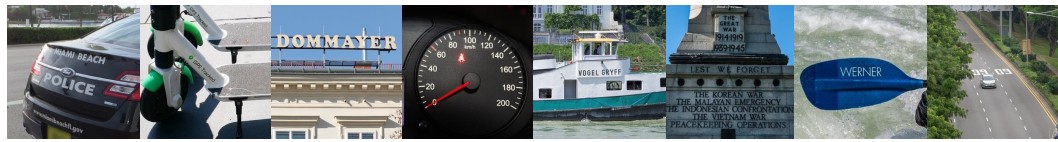

Figure 5: **Example images from our SA-Text.** Our dataset comprises high-quality, diverse images featuring text in varied sizes, styles, and layouts, including curved, rotated, and complex forms, providing a robust foundation for the proposed TAIR task.

### A.1    PIPELINE DETAILS

We apply our dataset curation pipeline to a subset of the SA-1B (Kirillov et al., 2023) dataset, a large-scale dataset originally designed for segmentation tasks. SA-1B (Kirillov et al., 2023) consists of 11M high-resolution images ($3,300 \times 4,950$ pixels on average) that have been downsampled so that their shortest side is $1,500$ pixels. This meets our requirements for high-quality images with sufficient resolution, suitable for the image restoration task. After processing with our dataset curation pipeline on a subset of SA-1B (Kirillov et al., 2023) (18.6% of the dataset), we are able to curate 100K high-quality crops that are densely annotated with text instances, enabling scalable dataset creation specifically for TAIR. Examples of the constructed SA-Text dataset are shown in Fig. 5. With its lightweight, modular design, the pipeline can be readily extended to other high-quality, large-scale datasets to supply additional data when required.

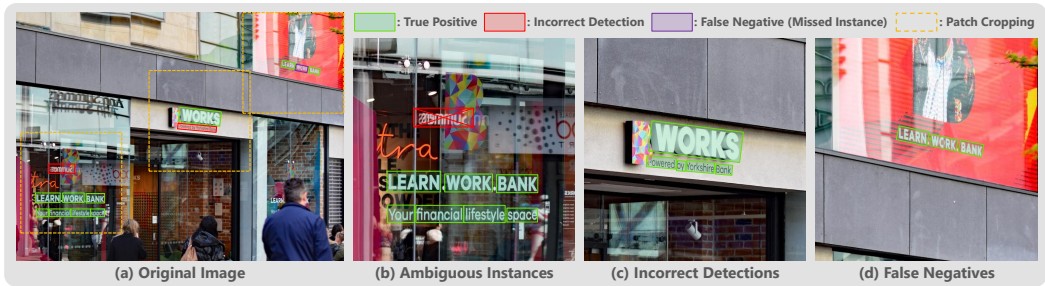

Figure 6: **Illustration of our dataset curation pipeline's effectiveness.** (a) Original high-resolution image with multiple text instances. (b) Ambiguous text instances are removed during the Vision-Language Model (VLM) filtering stage when the two VLMs produce differing recognition outputs. (c) Incorrect detections from the full image are corrected by re-running the detection model on smaller crops; here, the phrase "Powered by Yorkshire Bank" is successfully split into individual words. (d) False negatives (missed instances) from the initial detection on the entire image are effectively captured during the second detection pass on the smaller crops; the previously missed instance "WORK" is now correctly detected.

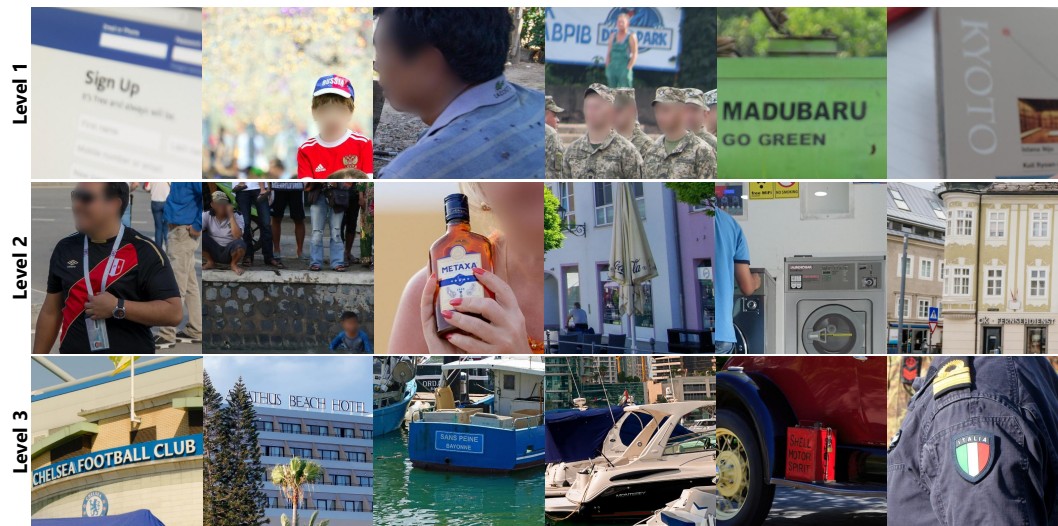

Figure 7: **Examples of images classified by blurriness.** Images are categorized into Levels 1-3: Level 1 (very blurry), Level 2 (slightly blurry), and Level 3 (clearly focused). Images classified as Level 1 and Level 2 are excluded from the final dataset to ensure that only clearly focused (Level 3) images are used for TAIR.

For the dedicated detection model used in the patch and box cropping stages, we utilize a state-of-the-art text-spotting model, DG-Bridge Spotter (Huang et al., 2024). As shown in Fig. 6, text instances initially missed by the detection model (false negatives) on full images are subsequently captured when the model is applied to smaller crops. We observe that running the detection model on smaller crops occasionally results in false positives, defined as detections of text in areas without actual text. Nonetheless, these instances are reliably filtered out in the subsequent VLM filtering stage, as at least one of the VLMs consistently identifies regions without text. Despite the strong recognition capabilities of DG-Bridge Spotter (Huang et al., 2024), we opt for VLMs due to their consistently superior recognition accuracy.

## A.2 VLM FILTERING DETAILS

For filtering out out-of-focus and intentionally blurred images, we employ a VLM (Bai et al., 2025) to classify each image's blurriness. We find that prompting the VLM to perform multiclass classification (as illustrated in Fig. 2) offers improved granularity and stricter filtering compared to using a simpler binary classification prompt ("blurry" vs. "not blurry"). Examples are shown in Fig. 7. After classification, crops labeled as Levels 1 and 2 (very blurry and slightly blurry, respectively) are filtered out from the final dataset. This VLM-based blur filtering ensures that only high-quality, sharply focused crops are used for TAIR.

## A.3 HUMAN VERIFICATION

| Method | Clean | Blurry | Intentionally Blurred |
|---|---|---|---|
| **Qwen-2.5-VL-7B (Ours)** | 71% | **88%** | **100%** |
| InternVL2.5-4B | **80%** | 61% | 90% |
| Laplacian variance (th=2000) | 50% | 88% | 88% |

Table 7: **Performance comparison of methods for blurry image filtering**

Our decision to use Qwen2.5-VL (Bai et al., 2025) for blur-filtering is based on a internal study we ran to evaluate the performance of different methods for assessing image blurriness. We compared the conventional Laplacian-variance baseline method against several VLMs and prompt formulations. The results are shown in Tab. 7, where the values represent the classification accuracy. For brevity, values for only the best-performing prompt are listed. We first constructed three balanced sets of

100 images each corresponding to "clean", "naturally blurry", and "intentionally blurred". When prompted with a short prompt that asks the VLM to classify sharpness on a 3-level scale, as in Fig. 2, Qwen2.5-VL achieved the scores above, exceeding both a classical Laplacian filter and InternVL2.5 (Chen et al., 2024b), in the two categories most critical for image restoration: detecting blur and privacy-related intentional masking. These results indicate that Qwen2.5-VL is sufficient for blur filtering even without any additional task-specific fine-tuning for image quality assessment.

Finally, to verify the robustness of our pipeline, we conducted a human audit. We randomly sampled 400 training images encompassing 911 text boxes and had coauthors annotate ground-truth labels. We subsequently compared these annotations against the outputs of our dataset curation pipeline, where we found only two instances where the recognition annotation differed from the ground truth (0.22% label noise). The only remaining issues stemmed from the detector missing tiny or heavily occluded text. This noise level is negligible for TAIR, and we consider the annotations sufficiently reliable for training and evaluation.

## A.4 FURTHER ANALYSIS

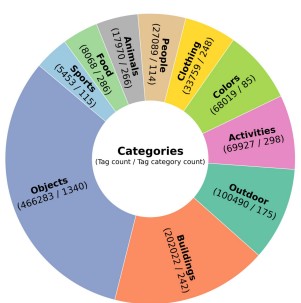

Figure 8: **Distribution of SA-Text.**

We further analyze SA-Text using RAM++ (Huang et al., 2023), which assigned tag categories to each image. This revealed 3,169 unique tags across the dataset as shown in Fig. 8. When categorized into 10 categories based on tags, the "Objects" category showed the largest proportion with over 466K instances, reflecting the prevalence of everyday objects. At the same time, other categories are also well represented, contributing to the diversity and balance of semantic domains covered in our dataset. This highlights the domain richness of our dataset, allowing it to capture both common and specialized concepts, and thereby supporting stronger generalization.

## A.5 EXTENSION TO MULTI-LINGUAL SETTINGS

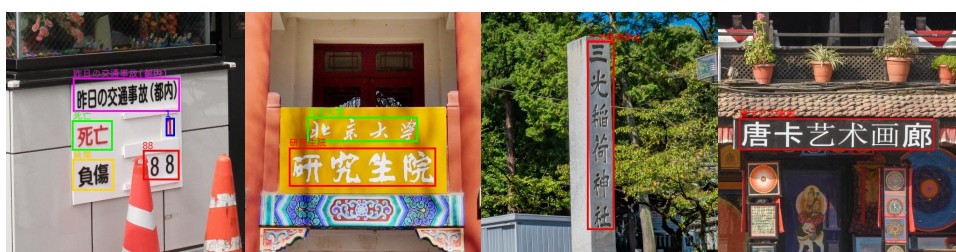

Figure 9: **Text annotation results on Chinese using our pipeline.**

Due to its modular and automated design, our dataset curation pipeline is inherently language-agnostic and can be easily adapted to multi-lingual settings. The language of detected text during the detection and cropping stages is determined by the text-spotting model. Although we used DG-Bridge Spotter (Huang et al., 2024), an English-focused model, for SA-Text, replacing it with a multilingual text spotter enables detection in other languages. The VLM-based recognition stage operates similarly, provided the models support the target language. As shown in Fig. 9, we apply the pipeline to Chinese text by substituting PP-OCRv5 (Cui et al., 2025) as the text spotter, while retaining Qwen2.5-VL (Bai et al., 2025) and OVIS2 (Lu et al., 2024) for recognition due to their strong Chinese language capability. To assess annotation quality, we conducted human verification on 100 images with Chinese annotations generated under this setup and observed quality comparable to SA-Text, with only one occlusion-related error. Ultimately, annotation quality depends on the underlying text-spotting and VLM components, and extending the pipeline to additional languages simply requires replacing these models with appropriate language-specific versions.

| Dataset Size | Backbone | Detection | | | End-to-End | |
|---|---|---|---|---|---|---|
| | | Precision(↑) | Recall(↑) | F1-Score(↑) | None(↑) | Full(↑) |
| 20K | ResNet-50 (He et al., 2016) | 77.67 | 65.91 | 71.31 | 1.81 | 1.96 |
| | Stable Diffusion (Rombach et al., 2022) | **82.26** | **81.78** | **82.02** | **27.42** | **43.87** |
| 100K | ResNet-50 (He et al., 2016) | 83.24 | 74.52 | 78.64 | 17.03 | 28.70 |
| | Stable Diffusion (Rombach et al., 2022) | **84.55** | **82.09** | **83.30** | **32.29** | **45.77** |

Table 8: **Comparsion of backbone features for training a text-spotting module (Zhang et al., 2022).** We evaluate the text detection and recognition performance of each trained model on SA-Text test set used in Tab. 2.

## B    DIFFUSION FEATURES FOR TEXT-SPOTTING

As mentioned in Sec. 4, we demonstrate that a text-spotting module can be trained directly using diffusion features for the text-spotting task (Zhang et al., 2022; Ye et al., 2023b; Qiao et al., 2024; Huang et al., 2024) instead of the commonly used ResNet (He et al., 2016) backbone, and that this approach not only enables effective training but also yields superior performance. To the best of our knowledge, this is the first work to demonstrate that diffusion features are suitable for effectively learning text-spotting.

### B.1    EXPERIMENTAL DETAILS

We train TESTR (Zhang et al., 2022) from scratch on subsets of SA-Text with a fixed timestep $t = 0$, varying both the vision features and the dataset size to evaluate their effects on text-spotting performance. We use ResNet-50 (He et al., 2016), which serves as the vision backbone in the original TESTR (Zhang et al., 2022), along with diffusion features extracted from Stable Diffusion 2.1[1] available on Hugging Face. We use model weights trained for 100K steps with a batch size of 8 and a learning rate of 1e-4, using 2 NVIDIA RTX 3090 GPUs. All other implementation details follow the default settings of TESTR (Zhang et al., 2022).

### B.2    RESULT ANALYSIS

As shown in Tab. 8, with a dataset size of 100K, diffusion-based features demonstrate superior recognition performance compared to ResNet (He et al., 2016) features. Importantly, when the dataset size is reduced to 20K samples, models relying on ResNet (He et al., 2016) features fail to effectively learn recognition, whereas diffusion features still achieve meaningful recognition learning even under limited data conditions. This shows that ResNet (He et al., 2016) features require a larger amount of training data to capture text semantics, which is consistent with the reliance on a large-scale synthetic dataset (Liu et al., 2020) in existing text-spotting methods (Zhang et al., 2022; Ye et al., 2023b; Qiao et al., 2024; Huang et al., 2024). In contrast, diffusion features, pretrained on diverse image-text pairs, can offer improved adaptability to text recognition, even when trained with limited real-world data.

## C    LOSS FUNCTIONS

**Stage 1.**    Stage 1 trains only the image restoration module using the following loss defined as:

$$\mathcal{L}_{\text{diff}} = \mathbb{E}_{z_0,\, t,\, p_t,\, c_t,\, \epsilon \sim \mathcal{N}(0,1)} \left[ \|\epsilon - \epsilon_\theta(z_t, t, p_t, c_t)\|_2^2 \right]. \tag{1}$$

**Stage 2.**    Stage 2 trains only the text-spotting module, while the restoration module remains frozen. Following transformer-based text-spotting methods (Huang et al., 2024; Zhang et al., 2022; Qiao et al., 2024), detection and recognition losses are computed via bipartite matching (Carion et al., 2020), solved using the Hungarian algorithm (Kuhn, 1955). Specifically, two separate loss functions

---

[1]`stabilityai/stable-diffusion-2-1-base`

are applied to the encoder and the dual decoder:

$$\mathcal{L}_{\text{enc}} = \sum_m \left( \lambda_{\text{cls}} \cdot \mathcal{L}_{\text{cls}}^{(m)} + \lambda_{\text{box}} \cdot \mathcal{L}_{\text{box}}^{(m)} + \lambda_{\text{gIoU}} \cdot \mathcal{L}_{\text{gIoU}}^{(m)} \right), \tag{2}$$

$$\mathcal{L}_{\text{dec}} = \sum_n \left( \lambda_{\text{cls}} \cdot \mathcal{L}_{\text{cls}}^{(n)} + \lambda_{\text{poly}} \cdot \mathcal{L}_{\text{poly}}^{(n)} + \lambda_{\text{char}} \cdot \mathcal{L}_{\text{char}}^{(n)} \right), \tag{3}$$

where $m$ and $n$ denote the numbers of instances with confidence scores exceeding a threshold $T$, and $\mathcal{L}_{\text{cls}}$, $\mathcal{L}_{\text{box}}$, $\mathcal{L}_{\text{gIoU}}$, $\mathcal{L}_{\text{poly}}$, and $\mathcal{L}_{\text{char}}$ denote the text classification loss, bounding box regression loss, generalized IoU loss (Rezatofighi et al., 2019), polygon regression loss (L1), and character recognition loss (cross-entropy), respectively. Each loss is weighted by its corresponding factor: $\lambda_{\text{cls}}$, $\lambda_{\text{box}}$, $\lambda_{\text{gIoU}}$, $\lambda_{\text{poly}}$, and $\lambda_{\text{char}}$.

**Stage 3.** In the final training stage, we optimize both the diffusion-based image restoration module and the text-spotting module. The total loss function, with a weight value $\lambda$, is formulated as follows:

$$\mathcal{L} = \mathcal{L}_{\text{diff}} + \lambda(\mathcal{L}_{\text{enc}} + \mathcal{L}_{\text{dec}}). \tag{4}$$

**Additional details.** Instance classification employs the focal loss (Lin et al., 2017), which is the difference of positive and negative terms to handle class imbalance. For the $j$-th query, the classification loss is given by

$$\begin{aligned} \mathcal{L}_{\text{cls}}^{(j)} = & -\mathbf{1}_{\{j \in \text{Im}(\sigma)\}} \, \alpha(1 - \hat{b}^{(j)})^\gamma \log(\hat{b}^{(j)}) \\ & -\mathbf{1}_{\{j \notin \text{Im}(\sigma)\}} \, (1 - \alpha)(\hat{b}^{(j)})^\gamma \log(1 - \hat{b}^{(j)}), \end{aligned}$$

where $\hat{b}^{(\cdot)}$ denotes the predicted confidence score, and $\text{Im}(\sigma)$ is the set of indices matched by the optimal bipartite assignment $\sigma$ between predictions and ground truth instances.

The control point regression loss encourages precise localization by minimizing the $\ell_1$ distance between predicted and ground truth control points:

$$\mathcal{L}_{\text{coord}}^{(j)} = \mathbf{1}_{\{j \in \text{Im}(\sigma)\}} \sum_{i=1}^{N} \left\| \hat{d}_i^{(j)} - d_i^{(\sigma^{-1}(j))} \right\|_1,$$

where $\hat{d}_i^{(\cdot)}$ and $d_i^{(\cdot)}$ denote predicted and ground truth control points, respectively, and $\sigma^{-1}$ maps prediction index $j$ to its assigned ground truth index.

Character classification is formulated as a multi-class problem and optimized via cross-entropy loss:

$$\mathcal{L}_{\text{char}}^{(j)} = \mathbf{1}_{\{j \in \text{Im}(\sigma)\}} \sum_{i=1}^{M} \left( -r_i^{(\sigma^{-1}(j))} \log \hat{r}_i^{(j)} \right),$$

where $\hat{r}_i^{(\cdot)}$ is the predicted probability for class $i$, and $r_i^{(\cdot)}$ is the one-hot encoded ground truth label.

## D  IMPLEMENTATION DETAILS

**Evaluation metrics.** We evaluate text-spotting performance (Liu et al., 2020; 2021; Zhang et al., 2022; Huang et al., 2024) using standard detection and recognition metrics. For detection, we report Precision (P), Recall (R), and F1-score (F), where a detection is considered correct if its Intersection over Union (IoU) with a ground-truth box exceeds 0.5. For recognition, we adopt two lexicon-based evaluation settings: None and Full. The None setting evaluates recognition without any lexicon, requiring exact matches to ground-truth transcriptions, reflecting performance in open-vocabulary scenarios. The Full setting permits matching predictions to the closest entry in a ground-truth lexicon, simulating closed-vocabulary conditions. This dual evaluation provides a comprehensive assessment of recognition accuracy.

| Deg. Level | Model | ABCNet v2 (Liu et al., 2021) | | | | | TESTR (Zhang et al., 2022) | | | | |
|---|---|---|---|---|---|---|---|---|---|---|---|
| | | Detection | | | End-to-End | | Detection | | | End-to-End | |
| | | Precision(↑) | Recall(↑) | F1-Score(↑) | None(↑) | Full(↑) | Precision(↑) | Recall(↑) | F1-Score(↑) | None(↑) | Full(↑) |
| Level1 | DiffBIR (Lin et al., 2024) | 76.29 | 56.44 | 64.88 | 23.14 | 32.73 | 84.00 | 52.13 | 64.34 | 25.51 | 35.47 |
| | DiffBIR† (Lin et al., 2024) | 53.01 | 51.86 | 52.43 | 15.26 | 20.71 | 60.53 | 51.99 | 55.94 | 16.78 | 22.82 |
| | **TeReDiff (Ours)** | **85.29** | **58.34** | **69.29** | **26.59** | **36.99** | **87.50** | **54.90** | **67.47** | **28.19** | **36.99** |
| Level2 | DiffBIR (Lin et al., 2024) | 72.96 | 53.94 | 62.03 | 19.60 | 27.52 | 79.69 | 50.50 | 61.82 | 21.64 | 30.36 |
| | DiffBIR† (Lin et al., 2024) | 53.28 | 51.18 | 52.21 | 14.75 | 19.61 | 58.85 | 50.18 | 54.17 | 16.15 | 21.43 |
| | **TeReDiff (Ours)** | **83.02** | **56.30** | **67.10** | **24.42** | **33.23** | **86.95** | **52.86** | **65.75** | **26.39** | **35.13** |
| Level3 | DiffBIR (Lin et al., 2024) | 59.30 | 42.20 | 49.31 | 13.88 | 19.39 | 72.27 | 38.98 | 50.65 | 15.61 | 22.67 |
| | DiffBIR† (Lin et al., 2024) | 44.16 | 40.62 | 42.31 | 10.15 | 14.45 | 49.73 | 41.25 | 45.09 | 10.75 | 15.41 |
| | **TeReDiff (Ours)** | **81.76** | **44.11** | **57.30** | **19.61** | **27.50** | **84.50** | **42.02** | **56.13** | **19.92** | **28.34** |

Table 9: **Text-spotting baseline comparison on SA-Text** Each block presents text restoration performance under varying degradation levels, evaluated using two text-spotting models (Liu et al., 2021; Zhang et al., 2022). 'None' indicates recognition without the use of a lexicon, while 'Full' denotes recognition assisted by a full lexicon. The best results are shown in **bold**, and the second-best are underlined.

| Model | ABCNet v2 (Liu et al., 2021) | | | | | TESTR (Zhang et al., 2022) | | | | |
|---|---|---|---|---|---|---|---|---|---|---|
| | Detection | | | End-to-End | | Detection | | | End-to-End | |
| | Precision(↑) | Recall(↑) | F1-Score(↑) | None(↑) | Full(↑) | Precision(↑) | Recall(↑) | F1-score(↑) | None(↑) | Full(↑) |
| DiffBIR (Lin et al., 2024) | 66.04 | 59.69 | 62.71 | 33.75 | 40.05 | 76.33 | 61.87 | 68.35 | 39.27 | 46.11 |
| DiffBIR† (Lin et al., 2024) | 55.31 | 56.02 | 55.67 | 26.85 | 31.23 | 58.99 | 60.19 | 59.58 | 31.41 | 35.98 |
| **TeReDiff (Ours)** | **83.95** | **67.58** | **74.88** | **48.39** | **55.01** | **84.30** | **67.37** | **74.89** | **49.39** | **56.45** |

Table 10: **Text-spotting baseline comparison on Real-Text.** We evaluate the text detection and recognition accuracy of the restored images on Real-Text using text-spotting models (Liu et al., 2021; Zhang et al., 2022). None' indicates recognition without the use of a lexicon, while 'Full' denotes recognition assisted by a full lexicon. The best results are shown in bold, and the second-best are underlined.

**Model overview.** Given a low-quality (LQ) image $I_{lq} \in \mathbb{R}^{H \times W \times 3}$, the objective is to recover a high-quality (HQ) image $I_{hq} \in \mathbb{R}^{H \times W \times 3}$ with enhanced visual and textual fidelity with $H = W = 512$. The LQ image is first processed by a lightweight degradation removal module (Liang et al., 2021) and encoded by a VAE (Kingma et al., 2013) to obtain a conditional latent $c \in \mathbb{R}^{\frac{H}{8} \times \frac{W}{8} \times 4}$. The HQ image is encoded and perturbed with noise to produce a noisy latent $z_t \in \mathbb{R}^{\frac{H}{8} \times \frac{W}{8} \times 4}$. These are channel-wise concatenated to form the input condition $c_t \in \mathbb{R}^{\frac{H}{8} \times \frac{W}{8} \times 8}$. Along with a diffusion timestep $t$ and a text prompt embedding $p_t \in \mathbb{R}^{n \times d}$, where $n = 77$ and $d = 1024$, $c_t$ is fed into the diffusion-based image restoration module. In our setting, $L = 9472$ and $D = 256$ are used for the multi-scale diffusion feature input $F \in \mathbb{R}^{L \times D}$. Deformable attention (Zhu et al., 2020) is employed to alleviate the high attention computation cost.

Following transformer-based text-spotting methods (Zhang et al., 2022; Qiao et al., 2024; Huang et al., 2024) and inspired by DETR (Carion et al., 2020), two sets of queries, $q_{det} \in \mathbb{R}^{Q \times D}$ and $q_{rec} \in \mathbb{R}^{Q \times D}$ with $Q = 100$, are provided to the detection and recognition decoders $\mathcal{D}^{det}$ and $\mathcal{D}^{rec}$, respectively. These queries are processed through cross-attention with the encoder output in the respectful decoder layers. The resulting predictions are denoted as $\{d^{(i)}\}_{i=1}^{K}$ and $\{r^{(i)}\}_{i=1}^{K}$, where $K$ is the number of instances with confidence scores above a threshold $T = 0.5$. Each $d^{(i)} = (d_1^{(i)}, \ldots, d_N^{(i)})$ represents a polygon with $N = 16$ control points, and $r^{(i)} = (r_1^{(i)}, \ldots, r_M^{(i)})$ contains $M = 25$ recognized characters.

**Training details.** To construct high-quality (HQ) and low-quality (LQ) training pairs for SA-Text, the LQ images were synthesized using the default degradation settings from the Real-ESRGAN pipeline (Wang et al., 2018). The final model used for performance evaluation in the main paper was trained on four NVIDIA H100 GPUs, with a batch size of 32 per GPU. Each of the three training stages (Stage 1 to Stage 3) was trained for 100,000 iterations. The hyperparameters for the text-spotting encoder loss function 2, decoder loss function 3, and stage3 loss function 4 are set as follows: $\lambda_{cls} = 2.0$, $\lambda_{coord} = 5.0$, $\lambda_{char} = 4.0$, $\lambda_{gIoU} = 2.0$ and $\lambda = 0.01$. During training, only the attention layers are updated in the image restoration module, whereas the text-spotting module undergoes full fine-tuning. The number of inference sampling steps for the diffusion based image restoration module was set to 50.

| Model | Text Condition | TESTR (Zhang et al., 2022) | | | | |
|---|---|---|---|---|---|---|
| | | Detection | | | End-to-End | |
| | | Precision(↑) | Recall(↑) | F1-Score(↑) | None(↑) | Full(↑) |
| Stage1 | Null | 82.09 | 50.27 | 62.36 | 25.36 | 31.66 |
| | Captioner (llava) | 83.84 | 52.67 | 64.70 | 27.39 | 35.58 |
| | Ground-truth | **86.79** | **63.42** | **73.28** | **33.68** | **44.21** |
| Stage3 | Null | 84.68 | 56.12 | 67.50 | 26.94 | 35.93 |
| | Captioner (Ours) | **87.50** | 54.90 | 67.47 | 28.19 | 36.99 |
| | Ground-truth | 86.49 | **63.28** | **73.09** | **34.71** | **44.87** |

(a) **SA-Text (Level 1).** Results evaluated on the SA-Text dataset with Level 1 degradation.

| Model | Text Condition | TESTR (Zhang et al., 2022) | | | | |
|---|---|---|---|---|---|---|
| | | Detection | | | End-to-End | |
| | | Precision(↑) | Recall(↑) | F1-Score(↑) | None(↑) | Full(↑) |
| Stage1 | Null | 81.77 | 47.37 | 59.99 | 21.24 | 29.79 |
| | Captioner (llava) | 82.01 | 49.82 | 61.99 | 24.76 | 31.70 |
| | Ground-truth | **85.09** | **61.56** | **71.44** | **32.51** | **42.71** |
| Stage3 | Null | 84.47 | 56.21 | 67.50 | 23.46 | 32.72 |
| | Captioner (Ours) | **86.95** | 52.86 | 65.75 | 26.39 | 35.13 |
| | Ground-truth | 86.18 | **61.60** | **71.85** | **33.31** | **43.40** |

(b) **SA-Text (Level 2).** Results evaluated on the SA-Text dataset with Level 2 degradation.

| Model | Text Condition | TESTR (Zhang et al., 2022) | | | | |
|---|---|---|---|---|---|---|
| | | Detection | | | End-to-End | |
| | | Precision(↑) | Recall(↑) | F1-Score(↑) | None(↑) | Full(↑) |
| Stage1 | Null | 75.69 | 39.53 | 51.94 | 16.32 | 22.22 |
| | Captioner (llava) | 75.74 | 41.61 | 53.72 | 18.02 | 24.46 |
| | Ground-truth | **80.60** | **55.94** | **66.04** | **27.08** | **37.57** |
| Stage3 | Null | 77.04 | 47.91 | 59.08 | 17.44 | 24.65 |
| | Captioner (Ours) | **84.50** | 42.02 | 56.13 | 19.92 | 28.34 |
| | Ground-truth | 80.04 | **55.44** | **65.51** | **27.91** | **37.76** |

(c) **SA-Text (Level 3).** Results evaluated on the SA-Text dataset with Level 3 degradation.

| Model | Text Condition | TESTR (Zhang et al., 2022) | | | | |
|---|---|---|---|---|---|---|
| | | Detection | | | End-to-End | |
| | | Precision(↑) | Recall(↑) | F1-Score(↑) | None(↑) | Full(↑) |
| Stage1 | Null | 81.91 | 62.68 | 71.02 | 44.11 | 50.70 |
| | Captioner (llava) | 81.23 | 65.48 | 72.51 | 44.72 | 51.28 |
| | Ground-truth | **83.46** | **73.61** | **78.23** | **52.59** | **59.28** |
| Stage3 | Null | 81.06 | 70.27 | 75.28 | 46.09 | 52.78 |
| | Captioner (Ours) | **84.30** | 67.37 | 74.89 | 49.39 | 56.45 |
| | Ground-truth | 83.41 | **75.28** | **79.14** | **54.06** | **60.74** |

(d) **Real-Text.** Results evaluated on the Real-Text dataset.

Table 11: **Additional ablations for SA-Text and Real-Text.** Extending training to Stage 3 enables the model to learn text-aware features, thereby improving text restoration performance. Moreover, conditioning with prompts from LLaVA (Stage 1) or our text-spotting module (Stage 3) yields further improvements compared to null prompts.

| Dataset | Model | PSNR(↑) | SSIM(↑) | LPIPS(↓) | DISTS(↓) | FID(↓) | NIQE(↓) | MANIQA(↑) | MUSIQ(↑) | CLIPIQA(↑) |
|---|---|---|---|---|---|---|---|---|---|---|
| SA-Texttest | Real-ESRGAN | 20.30 | **0.5967** | 0.3177 | 0.2116 | 56.68 | 4.626 | 0.5979 | 68.81 | 0.5245 |
| | SwinIR | 19.90 | 0.5852 | 0.3286 | 0.2159 | 57.41 | 4.790 | 0.5664 | 67.02 | 0.5045 |
| | ResShift | **20.64** | 0.5937 | 0.3509 | 0.2230 | 69.38 | 5.999 | 0.5525 | 64.52 | 0.4699 |
| | StableSR | 19.58 | 0.5103 | 0.3614 | 0.2217 | 49.34 | **4.418** | 0.6036 | 66.88 | 0.5077 |
| | DiffBIR | 19.58 | 0.4965 | 0.3636 | 0.2086 | 45.10 | 5.107 | 0.6771 | 73.33 | 0.6589 |
| | DiffBIR† | 16.81 | 0.4638 | 0.4001 | 0.2219 | 47.28 | 5.449 | 0.6890 | 72.55 | 0.6345 |
| | SeeSR | 19.72 | 0.5671 | 0.3173 | 0.1943 | 40.14 | 5.050 | 0.6730 | **73.57** | 0.6490 |
| | SUPIR | 16.81 | 0.4084 | 0.4723 | 0.2554 | 56.05 | 5.626 | 0.6723 | 70.10 | 0.5980 |
| | FaithDiff | 19.78 | 0.5444 | 0.3214 | 0.1966 | 39.09 | 4.936 | **0.7075** | 73.51 | **0.6630** |
| | **TeReDiff (Ours)** | 19.71 | 0.5717 | **0.2828** | **0.1702** | **36.94** | 5.452 | 0.6471 | 72.07 | 0.6145 |
| Real-Text | Real-ESRGAN | 23.84 | 0.8062 | 0.2378 | **0.2040** | 68.72 | 6.356 | 0.5437 | 58.72 | 0.3907 |
| | SwinIR | **24.31** | **0.8122** | **0.2314** | 0.2049 | 67.19 | 6.799 | 0.4905 | 55.39 | 0.3609 |
| | ResShift | 24.29 | 0.7989 | 0.2936 | 0.2559 | 85.94 | 7.977 | 0.4940 | 56.56 | 0.4023 |
| | StableSR | 24.20 | 0.7745 | 0.2744 | 0.2225 | **58.81** | 7.107 | 0.5394 | 57.85 | 0.3821 |
| | DiffBIR | 22.70 | 0.6365 | 0.4090 | 0.2849 | 81.56 | 6.842 | 0.6194 | 67.28 | **0.5748** |
| | DiffBIR† | 18.92 | 0.5042 | 0.5168 | 0.3058 | 75.78 | 8.126 | 0.5937 | 62.12 | 0.5053 |
| | SeeSR | 23.63 | 0.7858 | 0.2852 | 0.2477 | 79.75 | 7.259 | 0.6001 | 67.52 | 0.5432 |
| | SUPIR | 16.64 | 0.3483 | 0.6405 | 0.3579 | 99.13 | 14.20 | 0.6255 | 63.13 | 0.4842 |
| | FaithDiff | 24.02 | 0.7711 | 0.2981 | 0.2454 | 67.15 | 6.869 | **0.6293** | 65.19 | 0.5350 |
| | **TeReDiff (Ours)** | 23.37 | 0.7849 | 0.2848 | 0.2386 | 68.94 | 7.643 | 0.5637 | 62.02 | 0.4545 |

Table 12: **Quantitative results of image restoration.** We evaluate the image quality of our TeReDiff compared with other baseline models. The degradation pipeline (Wang et al., 2021) used in prior works is applied. DiffBIR (Lin et al., 2024)† (denoted as v2.1 in Fig. 1) uses GitHub-released weights that are further trained from those in the original paper.

# E ADDITIONAL QUANTITATIVE RESULTS

## E.1 EXTENDED IMAGE QUALITY COMPARISON

Tab. 12 shows the additional results for all baseline models on image restoration metrics. It is clear that our proposed task, TAIR, does not negatively affect overall image restoration performance, while improving upon the model's ability to accurately restore text. We posit that these image restoration metrics are not an ideal measure for evaluating text fidelity or legibility, which is the main focus of this paper.

## E.2    Extended Baseline Comparison

Our main baseline image restoration model is DiffBIR (Lin et al., 2024). We further include comparisons with DiffBIR$^{\dagger}$ (denoted as DiffBIR v2.1), which leverages publicly available weights from the official GitHub repository that have undergone additional fine-tuning beyond what was reported in the original publication. The comparative results on text restoration performance are provided in Tab. 9 for SA-Text and Tab. 10 for Real-Text.

## E.3    Extended Ablation Experiments

Extending the analysis in Tab. 6 of the main paper, we present additional ablation results in Tab. 11, including evaluations across the three degradation levels on SA-Text and on the Real-Text dataset. The table compares (1) two model training stages, referred to as Stage1 and Stage3, and (2) three prompting strategies for the image restoration module: Null (no prompt), Captioner (LLaVA or our text-spotting module), and Ground-truth (ground truth text). The ground truth prompt is constructed from the actual textual content present in the LQ image, whereas prompts of 'Captioner' use recognized texts extracted from the LQ image by our text-spotting module or a conventionally used LLaVA (Liu et al., 2023b).

**Overall comparison.**    Comparing Stage 1 and Stage 3 under Null prompt text conditioning, we observe accuracy gains of +4.27%, +2.93%, +2.43%, and +2.08% in Tab. 11a through Tab. 11d, respectively, indicating the benefit of training with text-aware supervision in Stage 3. When using a captioner, we employ the LLaVA (Liu et al., 2023b) captioner for Stage 1 due to the absence of a trained text-spotting module, whereas Stage 3 leverages our trained text-spotting module as the captioner. The improved performance of Stage 3 demonstrates that our module provides more accurate and text-aware prompts, enhancing restoration quality. Finally, when utilizing ground-truth texts from LQ images for both stages, substantial performance improvements are observed with minimal differences between them. The ideal scenario in which the restoration module is provided with exact textual information is not achievable in practice and primarily serves to establish an upper bound on performance.

**Comparison on SA-Text degradation levels.**    Comparing Stage1$_{\mathrm{pr}}$ and Stage3$_{\mathrm{pr}}$ across the three degradation levels of SA-Text, we observe that the performance gap increases with higher degradation severity. This highlights the importance of Stage3, which learns text-aware features and enables prompting the restoration module using our text-spotting module, proving these prompts' superiority over those from an external LLaVA captioner.

## E.4    User Study Evaluation

| Criteria | DiffBIR | Ours |
|---|---|---|
| Text Quality | 2.20% | **97.8%** |
| Image Quality | 9.2% | **90.8%** |

Table 13: **User study on text and image restoration quality.**

To evaluate the quality of both text and image restoration achieved by our TeReDiff, we conducted a simple user study comparing it with our baseline, DiffBIR (Lin et al., 2024). The study included 10 samples: 5 from SA-Text Level 3 and 5 from Real-Text. A total of 50 participants took part in the evaluation. Samples for SA-Text and Real-Text were selected from the examples shown in Tab. 2 and Tab. 3, respectively. As shown in Tab. 13, the user study results indicate that our TeReDiff outperforms the baseline in both text restoration and visual quality. These results highlight that humans often consider text semantics when evaluating image quality, an aspect not fully captured by existing image metrics.

Each participant evaluated the samples based on the following set of questions.:

1. Which image better restores the text content? (Image 1 / Image 2)
2. Which image better restores the overall appearance? (Image 1 / Image 2)

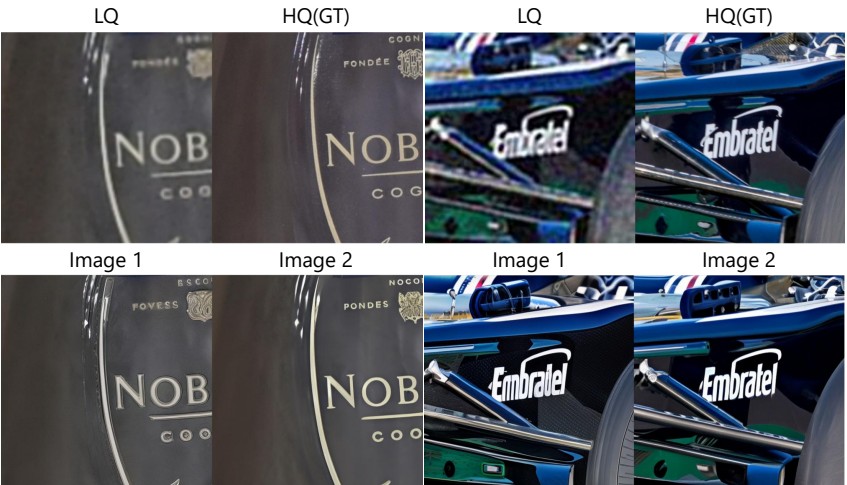

Figure 10: **Example samples for user study.**

## F    ADDITIONAL QUALITATIVE RESULTS

In Fig. 11, Fig. 12, Fig. 13, Fig. 14, and Fig. 15 we show further qualitative results on text-aware image restoration (TAIR). The results on SA-Text across different degradation levels and those on Real-Text, demonstrate that our model outperforms other diffusion-based methods.

Fig. 16 shows the restoration results of TeReDiff on three difficulty levels (easy, medium, hard) from TextZoom (Wang et al., 2020). TeReDiff demonstrates strong text restoration performance, producing results that are even clearer than the ground-truth high-resolution images.

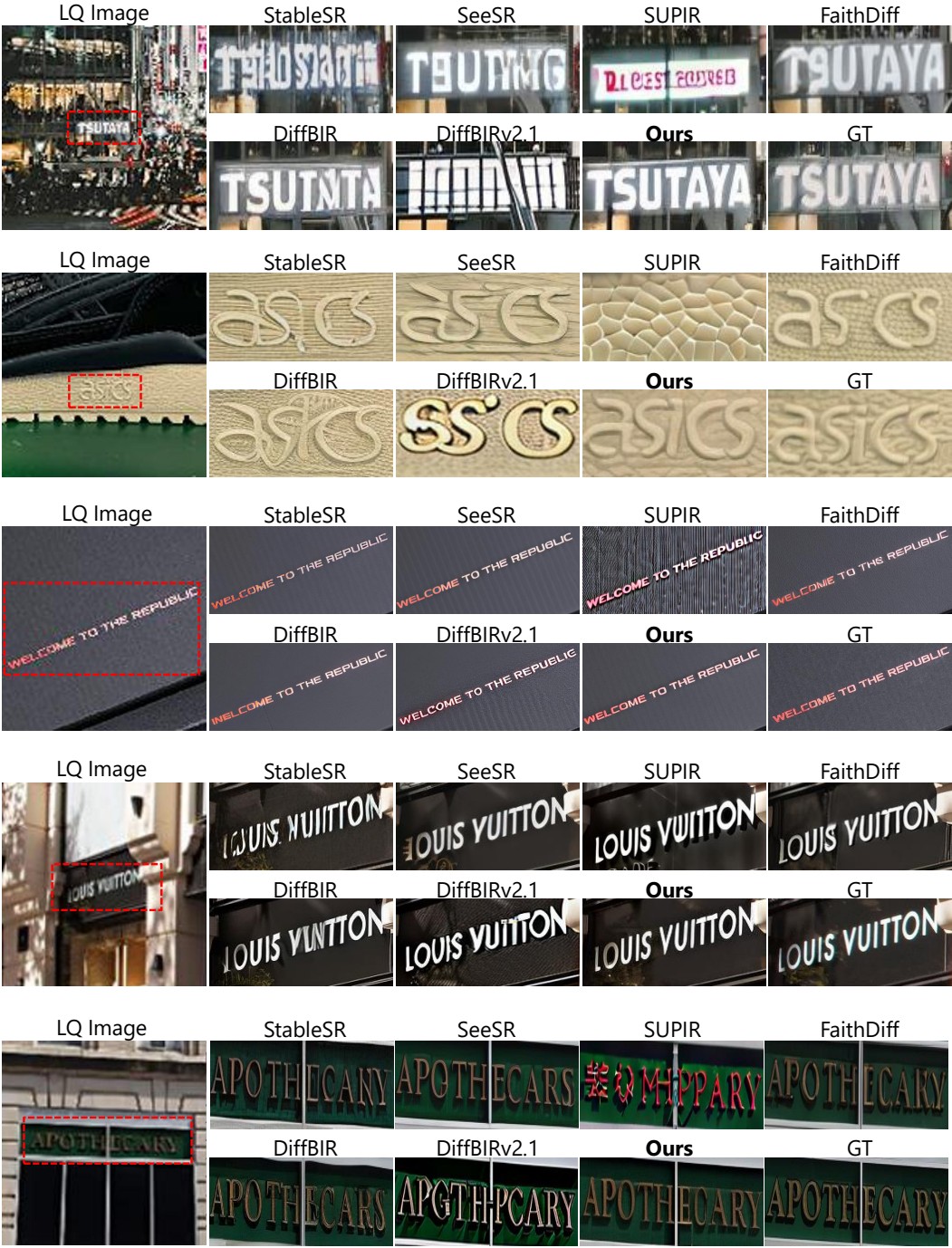

Figure 11: **Qualitative results on SA-Text test set Level 1.**

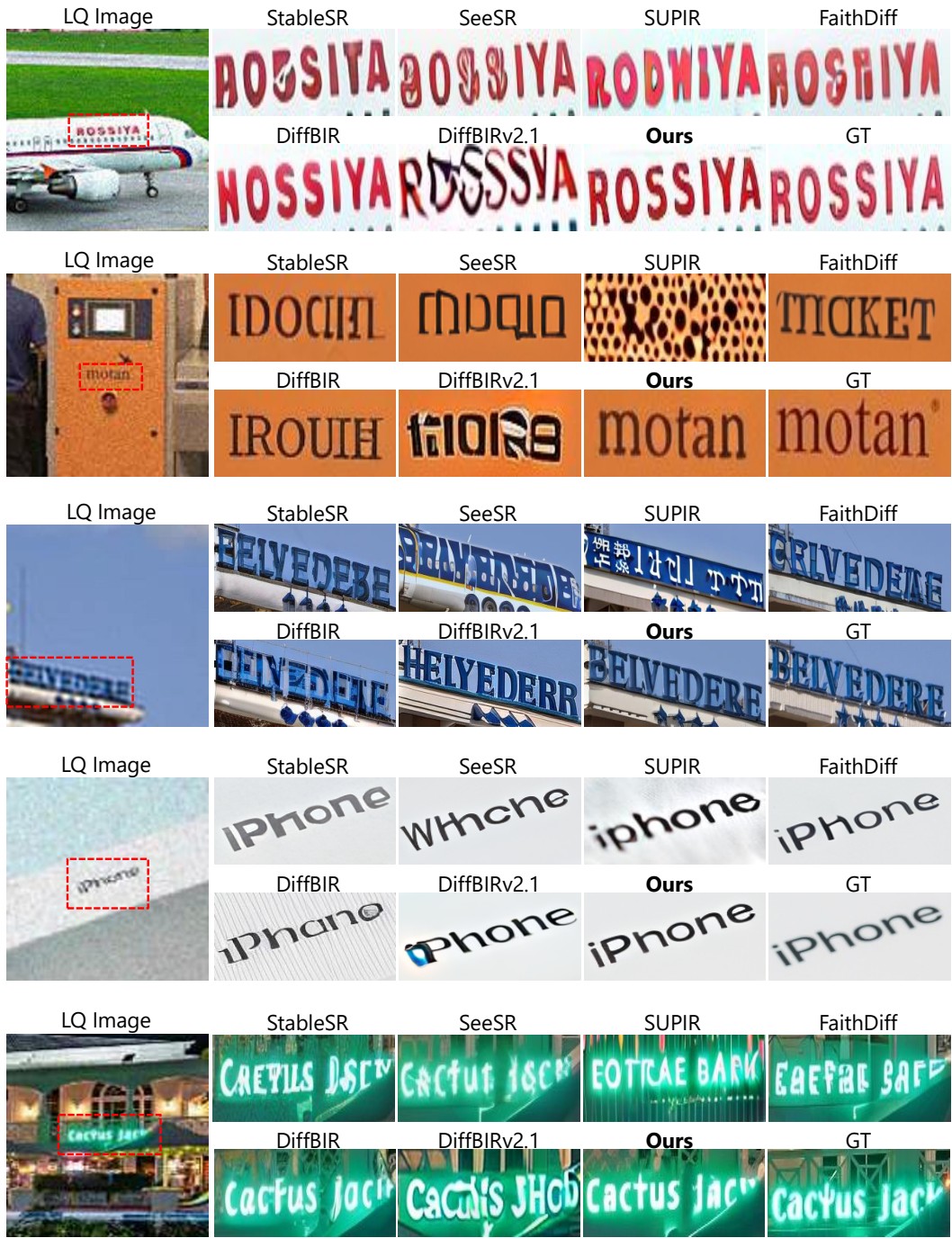

Figure 12: **Qualitative results on SA-Text test set Level 2.**

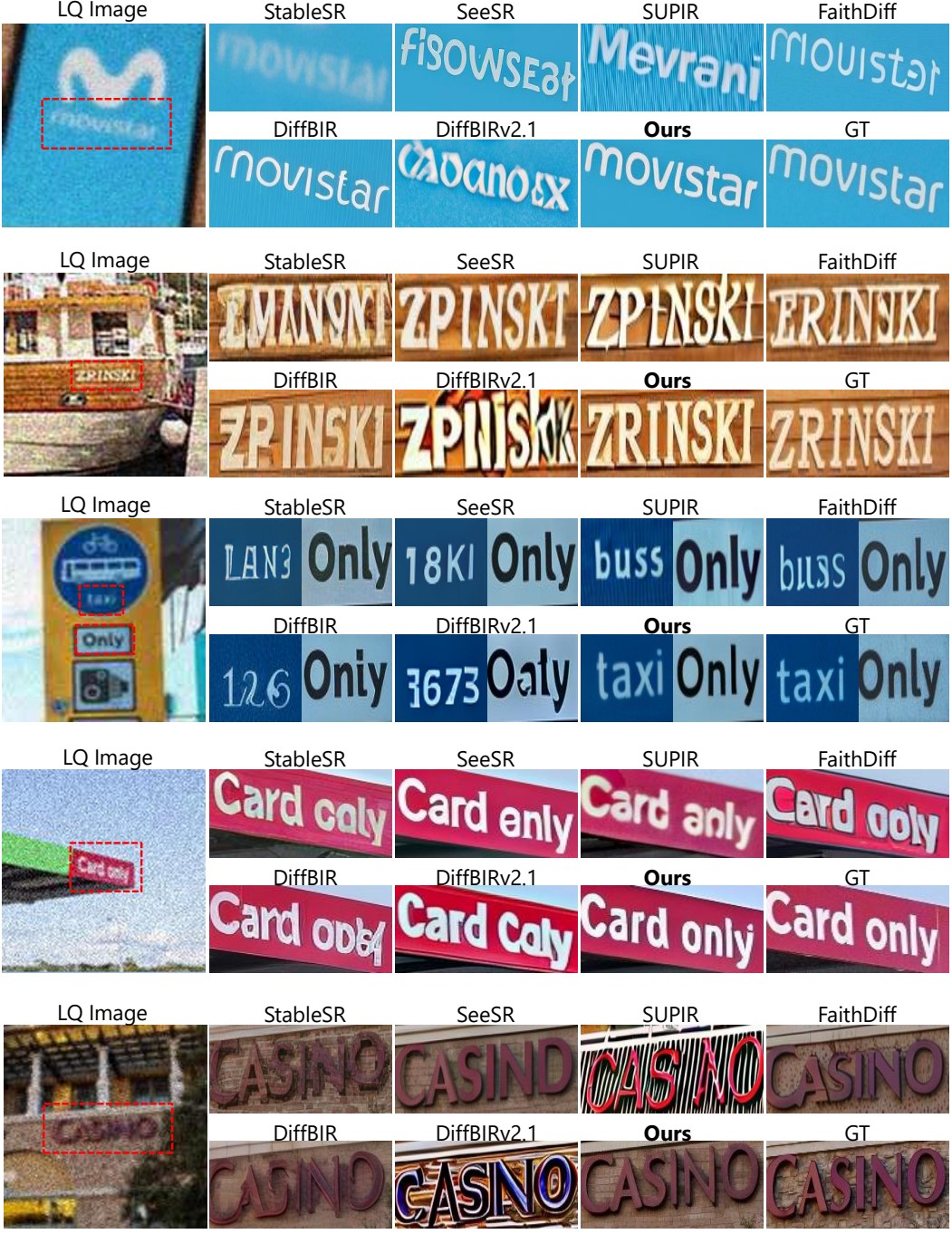

Figure 13: **Qualitative results on SA-Text test set Level 3.**

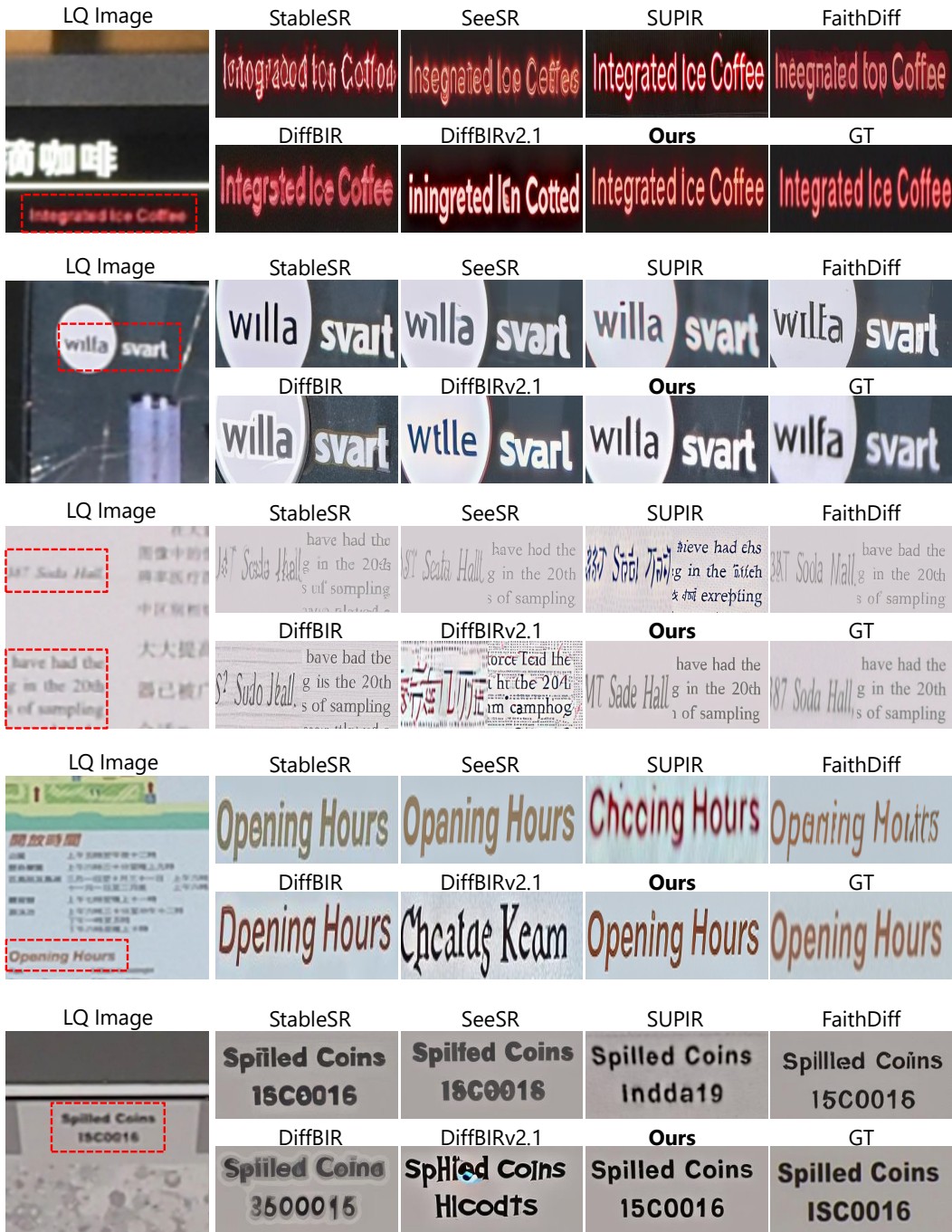

Figure 14: **Qualitative results on Real-Text (1/2).**

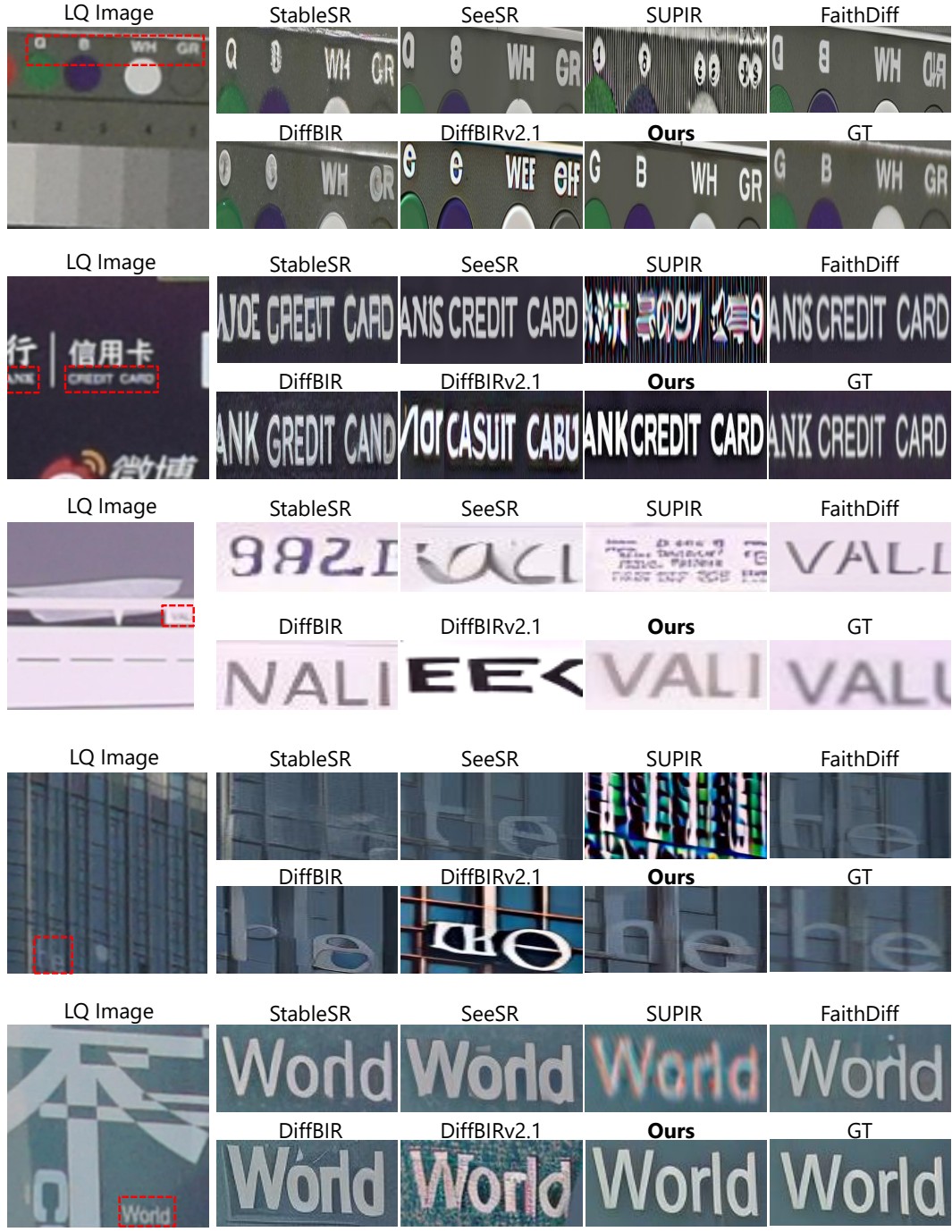

Figure 15: **Qualitative results on Real-Text (2/2).**

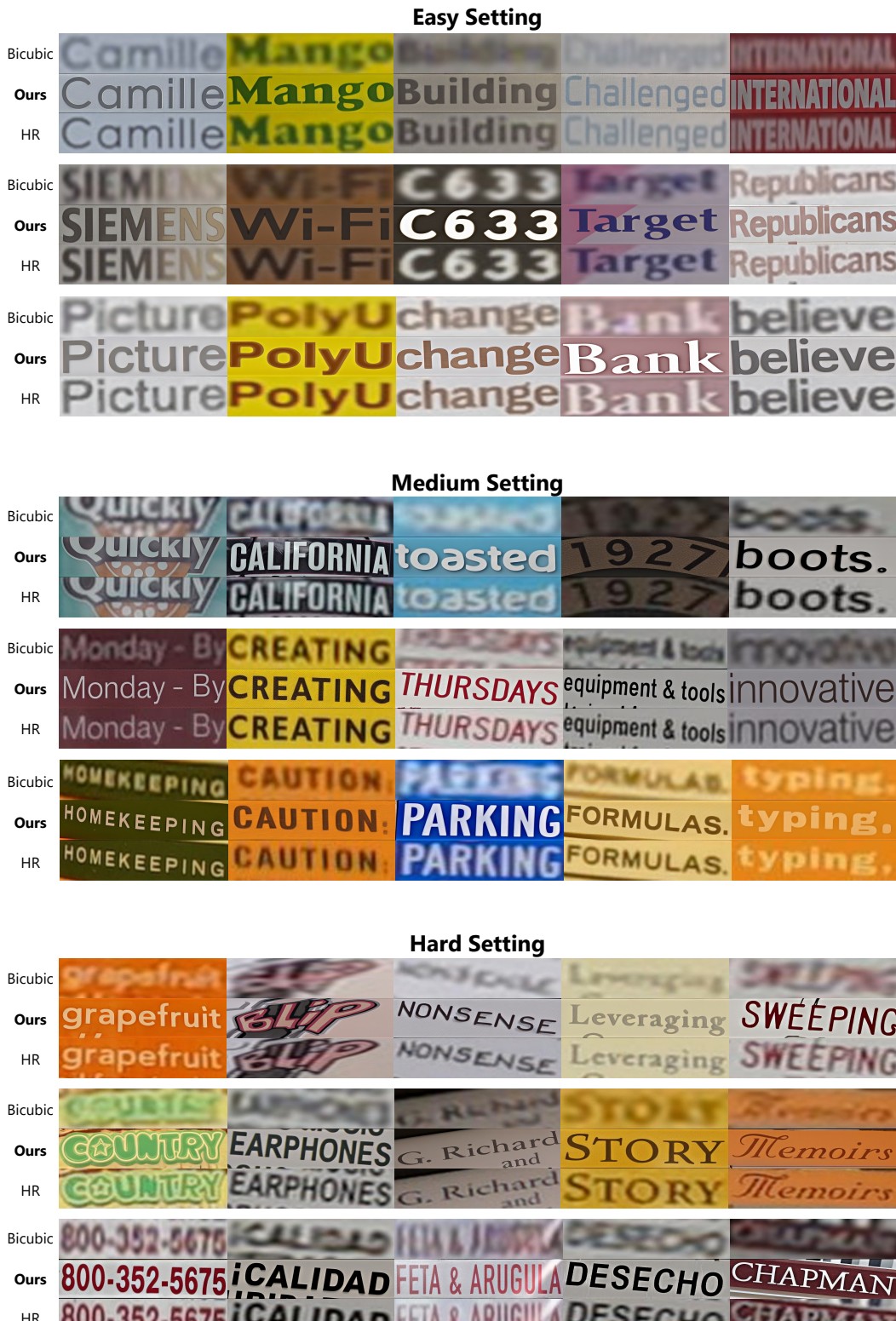

Figure 16: **Qualitative results on three difficulty settings of TextZoom.**

# G DISCUSSION

## G.1 VLM CORRECTION

| Model | ABCNet v2 (Liu et al., 2021) | | | | | TESTR (Zhang et al., 2022) | | | | |
| | Detection | | | End-to-End | | Detection | | | End-to-End | |
| | Precision(↑) | Recall(↑) | F1-Score(↑) | None(↑) | Full(↑) | Precision(↑) | Recall(↑) | F1-Score(↑) | None(↑) | Full(↑) |
|---|---|---|---|---|---|---|---|---|---|---|
| TeReDiff (w/o VLM) | **83.95** | 67.58 | 74.88 | 48.39 | 55.01 | 84.30 | 67.37 | 74.89 | 49.39 | 56.45 |
| TeReDiff (w/ VLM) | 83.70 | **68.28** | **75.21** | **49.70** | **55.99** | **85.87** | **67.74** | **75.74** | **50.87** | **57.25** |

Table 14: **VLM correction results on Real-Text.** Employing VLM (Bai et al., 2025) to refine the text predictions generated by the text-spotting module further improves text restoration performance.

Providing the diffusion-based image restoration module with accurate textual prompts is essential for achieving faithful and precise text restoration. Although our TeReDiff framework incorporates a text-spotting module to generate such prompts and has demonstrated effectiveness, erroneous text predictions may cause the restoration module to reconstruct incorrect textual content. To address this limitation, we further investigate the use of a vision-language model (VLM) to refine the predictions of the text-spotting module. Since the spotting module lacks the ability to fully capture character-level dependencies and contextual relationships within the predicted text, we leverage the linguistic knowledge of VLMs to correct these predictions. The quantitative restoration results obtained using the 3B Qwen2.5-VL (Bai et al., 2025) to correct text predictions on Real-Text are reported in Tab. 14.

For the VLM, two inputs are provided:

1. **Visual input:** The restored image at each denoising timestep.

2. **Textual input:**

```
The image contains degraded or low-quality text.  The
OCR-predicted text may contain errors.  Use both the
visual appearance of the text and the predicted text to
infer and correct the actual text.  Only recognize and
correct English text.

OCR prediction:  "[Text prediction from the TS module]"

Return only the corrected English text from the image.
```

## G.2 COMPUTATIONAL COMPLEXITY AND INFERENCE TIME

| Criteria | TeReDiff (Ours) | DiffBIR (Baseline) | SeeSR | StableSR | FaithDiff |
|---|---|---|---|---|---|
| Model Size (M) | 1706.20 | 1666.75 | 2283.70 | 1409.11 | 2612.20 |
| Inference Time (s) | 7.29 | 6.59 | 7.24 | 11.18 | 9.97 |

Table 15: **Comparison of model sizes and inference times across different methods.** Experiments were conducted on an NVIDIA RTX 3090 GPU.

TeReDiff has a total of $1,706.2M$ parameters, comprising $1,682.54M$ in the image restoration module and the remaining $23.65M$ in the text-spotting module, which is an increase of only $\sim1.4\%$ over the backbone alone. Despite the addition of a text-spotting module to an existing restoration module, this leads to a negligible increase in computational overhead. As a result, TeReDiff attains model size and inference time on par with (and sometimes better than) restoration-only baselines of similar or larger capacity. As shown in Tab. 15, TeReDiff is smaller and/or faster than all compared methods except the backbone baseline on which it is built, while delivering substantially better text restoration. This demonstrates that TeReDiff improves text-aware image restoration at minimal computational cost, underscoring its efficiency and real-world applicability.

## G.3 ANALYSIS OF FAILURE CASES

To further investigate the limitations of our method, we analyzed the "None" results from Tab 2's TESTR (Zhang et al., 2022) evaluation and report the minimum text sizes in Tab. 16. Specifically,

Tab. 16 shows the minimum bounding box sizes of text that was either correctly recognized by the text spotting module at the initial denoising step (Case 1) or accurately restored by TeReDiff (Case 2).

|        | SA-Text (Lv1) | SA-Text (Lv2) | SA-Text (Lv3) | Real-Text |
|--------|---------------|---------------|---------------|-----------|
| Case 1 | 16×17         | 16×17         | 28×22         | 20×16     |
| Case 2 | 14×27         | 25×20         | 35×26         | 20×17     |

Table 16: **Minimum text box sizes (HxW) for accurate text restoration.** Evaluated on 512×512 images from SA-Text and Real-Text. Case 1 denotes text detection outputs of the Text Spotting Module, while Case 2 denotes the text restoration results based on the E2E (None) F1-score of TESTR (Zhang et al., 2022).

We statistically observed a tendency for smaller text to be either more difficult to spot (Case 1) or harder to successfully restore (Case 2). However, various types of noise are randomly applied by the degradation kernel from Real-ESRGAN (Wang et al., 2021), and other additional factors, such as the specific font and text length, may also affect the restoration results. The experiments in Tab. 16 help identify the practical boundaries of text size at which the text spotting module fails to detect text and the restoration module fails to recover it under various degradation scenarios.

## H USAGE OF LARGE LANGUAGE MODELS

Following the ICLR 2026 submission policy, we disclose that the use of Large Language Models aids in correcting grammar and editing LaTeX syntax.

