# OpenReview forum: "Text-Aware Image Restoration with Diffusion Models"
_ICLR.cc/2026/Conference — ICLR 2026 Poster_

### Official Review · Reviewer_cLfC · 2025-10-25

**Soundness:** 3
**Presentation:** 3
**Contribution:** 2
**Rating:** 4
**Confidence:** 4

**Summary:**

This paper introduced the Text-Aware Image Restoration (TAIR) task, which requires simultaneous recovery of visual content and textual fidelity. And proposed corresponding diffusion-based method TeReDiff, which leverages text-spotting task to supervise intermidiate features and guide the optimization process. Also the paper built high-quality scene images SA-Text benchmark with dense annotations of diverse and complex text instances. Extensive experiments demonstrate the effectiveness of the proposed method.

**Strengths:**

Strengthens:
1. Proposed an exhaustive pipeline for building the dataset and made the large-scale high-quality scene images SA-Text benchmark, contributing to the advancement of research community.
2. Introduced text-spotting module to supervision on intermediate features, explicitly guides optimization and empowers the model to implicitly perceive textual information from LQ inputs.
3. The proposed method achieves outstanding performance on TAIR, surpassing the existing Real-SR methods.

**Weaknesses:**

Weaknesses:
1. The novelty is limited.
    - The proposed method integrates a plain ControlNet to inject LQ priors, without any specific design tailored for the text-focused nature of the task.
    - The detail structure of Text-spotting module like optimization objective also has not been fully described, which undermines the reproducibility.
    - This paper is not the first to define the TAIR task, as earlier work [1] have already proposed similar tasks.
    References:
    [1]Q. Hu, L. Fan, Y. Luo, Y. Yu, X. Guo, and Q. Fan, “Text-Aware Real-World Image Super-Resolution via Diffusion Model with Joint Segmentation Decoders,” CoRR, vol. abs/2506.04641, 2025, doi: 10.48550/ARXIV.2506.04641.
2. The experiments are insufficient.
    - TAIR focuses on both visual content restoration and textual fidelity. However, the paper lacks comprehensive evaluations on real-world image super-resolution tasks. Moreover, the metrics individually assess either scene reconstruction quality or text restoration quality, and thus fail to reflect the balance between these two aspects.
    - While the visual results in the paper are impressive, the examples contain only a single line of text with limited style variation, which makes it difficult to demonstrate the effectiveness of the proposed method in real-world scenarios with diverse text densities and styles.
3. The layout is unsatisfactory. The gap between the text and corresponding tables/figures is too large, making readers hard to follow the content, e.g. Figure 3, Table 2.

**Questions:**

1. Does the proposed method only support English text? How does it perform on non-Latin characters?

2. Would the optimization objectives of the TAIR task affect the restoration of images without text, for example, causing tree branches to be mistakenly reconstructed as characters?

---

> ### Author Response · Authors · 2025-11-26
>
> Dear Reviewer cLfC,
>
> We sincerely appreciate your insightful comments and questions, which have greatly contributed to improving our work. We address your concerns as follows.
>
> ---
>
> > **[W1 - (1),(3)] About contribution and not a first work**
>
> [A1] Thank you for these comments. We would like to clarify our contributions:
>
> Our contribution is not in novel architecture design, but in proposing a text-aware restoration framework with a text-spotting module. Specifically, we are the first to train a text-spotting module using diffusion features and integrate its OCR outputs as prompts during denoising. We provide comprehensive analysis of this approach in Appendix B.
>
> While we consider [1] to be concurrent work, we will tone down our task novelty claims in the revision. We would like to clarify that our task definition differs from [1] in a key aspect: while [1] uses segmentation maps for text-aware restoration, our task is formulated around leveraging OCR outputs from the text-spotting module. This formulation represents a distinct, linguistically-grounded approach to text-aware image restoration that enables actual text recognition, unlike the segmentation-based approach in [1]. We will add the citation in the revision.
>
> > **[W1 - (2)] Detail about text-spotting module**
>
> [A2] Thank you for pointing this out. Appendix C and D provide comprehensive details on the text-spotting module, including the network architecture, optimization objectives, model parameters, and loss function weights.
>
> > **[W2 - (1)] Results for real-world degradations & About metrics**
>
> [A3] As outlined in Section 5.1 in the main paper, we already provide the evaluation of our method on two real-world datasets: Real-Text and TextZoom. The Real-Text dataset is constructed from real-world HR–LR image pairs derived from RealSR and DRealSR, whereas TextZoom comprises real-world cropped image pairs commonly used in scene text image super-resolution (STISR). Quantitative restoration results for these datasets are presented in Tab. 3 and Tab. 5 of the main paper. Complementary qualitative comparisons are provided in Fig. 13 and Fig.14 from Appendix, further highlighting the effectiveness of our approach in restoring text from real-world degraded inputs.
>
> Furthermore, we would like to clarify our evaluation approach. Our primary objective is not to balance scene and text reconstruction equally, but rather to enable effective text restoration through joint training of a text-spotting module with diffusion features for text-aware image restoration. Accordingly, our evaluation focuses primarily on text restoration metrics to assess whether our method successfully restores text. We additionally report image quality metrics to demonstrate that our text-focused approach does not compromise overall image restoration performance. This evaluation strategy directly reflects our goal: achieving strong text restoration while maintaining competitive general image quality.
>
> > **[W2 - (2)] Diverse texts scenarios**
>
> [A3] We would like to clarify that our method handles diverse text styles as already shown in  Figs 10-13. While we show single-line examples for better visualization, our method effectively handles multi-line text as well. This can be seen in each third row of Fig. 12 and 13, and the fifth row of Fig. 13, where multiple lines of text are successfully restored.
>
> > **[W3] Layout**
>
> [A4] Thank you for pointing this out. We acknowledge the layout issues and will adjust the placement of figures and tables in the revision for better readability.
>
> > **[Q1] About language-agnostic**
>
> [A5] Thank you for raising this important point regarding language generality.
>
> Our SA-Text annotation pipeline is language-agnostic by design, as it leverages VLM's multilingual capabilities for automatic annotation. During development, we observed the VLM successfully handling complex scripts like Chinese, demonstrating the pipeline's potential for multilingual dataset construction.
>
> While our dataset curation pipeline can support multiple languages, we trained the text spotting module (TSM) exclusively on English. This decision was made to maintain focus on our core contribution. As the first work to learn a TSM from diffusion features, we prioritized validating this novel learning paradigm with English to ensure clear, interpretable results, rather than addressing the substantial complexities of multilingual text spotting [2].
>
> Training separate TSMs for multiple languages would require individual models, datasets, and extensive experiments for each language, valuable but orthogonal to our core contribution. Our method is fundamentally language-agnostic and provides a solid foundation for future multilingual extensions.

---

> ### Author Response · Authors · 2025-11-26
>
> > **[Q2] Affect of optimization objectives**
>
> [A6] Thank you for this important question. We can address this concern through our ablation studies in Tab. 6 (a). If our model mistakenly reconstructed non-text regions (e.g., tree branches) as text, we would expect to see lower precision in text detection metrics. However, when comparing Stage 1 (diffusion loss only) with Stage 3 (joint optimization with text-spotting objectives), we observe improvements in both precision and recall for text detection. This indicates that our joint optimization enhances text restoration without introducing false text artifacts in non-text regions.
>
> Additionally, our inference pipeline incorporates a safeguard against such cases. The text-spotting module outputs confidence scores alongside detection results, and we apply confidence thresholding to use only high-confidence predictions for prompt guidance (details in Appendix D). This mechanism helps prevent potential false text reconstruction.
>
> ---
>
> Many thanks to Reviewer cLfC for the valuable feedback. We have carefully addressed each comment and hope our responses resolve your concerns. Your comments greatly improve the clarity and quality of our work, and we appreciate your time and effort. We’re happy to address any further concerns during the discussion period.
>
> ## **Reference**
> [1] Hu, Qiming, et al. "Text-Aware Real-World Image Super-Resolution via Diffusion Model with Joint Segmentation Decoders." arXiv preprint arXiv:2506.04641 (2025).
>
> [2] Huang, Jing, et al. "A multiplexed network for end-to-end, multilingual OCR." Proceedings of the IEEE/CVF conference on computer vision and pattern recognition. 2021.

---

### Official Review · Reviewer_5iBr · 2025-10-27

**Soundness:** 2
**Presentation:** 3
**Contribution:** 3
**Rating:** 6
**Confidence:** 4

**Summary:**

This paper addresses the critical challenge of text-image hallucination in diffusion models for image restoration, proposing a novel task called Text-Aware Image Restoration (TAIR). To advance this field, the authors constructed a large-scale benchmark with dense text annotations named SA-Text dataset. The key innovation is the TeReDiff model, where the joint training of a text-spotting module with the restoration module enables text-aware supervision during restoration, achieved new state-of-the-art results on TextZoom and demonstrated superior super-resolution results.

**Strengths:**

1. Excellent originality: This paper purposefully introduces a new task (TAIR), along with a corresponding dataset and a SOTA model capable of performing both tasks simultaneously.
2. Good quality and performance: The paper constructs a large-scale, high-quality dataset for TAIR and designs a novel model architecture that leverages joint training for both text-spotting and restoration tasks, achieving outstanding performance across multiple benchmarks.
3. The paper is well-written and easy to follow.

**Weaknesses:**

1. While the paper emphasizes that the new TAIR task differs from previous models by focusing on text-image hallucination (text readability), it fails to sufficiently demonstrate the model's superiority in this core aspect. The evidence is limited to a few comparative images and existing metrics that are not fully relevant. This lack of qualitative comparison methods specific to the TAIR task undermines the credibility of its performance evaluation.
2. The comparison in Table 4 is insufficient to substantiate the claim of superior image quality, as it includes too few diffusion-based SR models. To properly validate performance, the evaluation should be expanded to include more competing methods, particularly established models like StableSR and ResShift that are already discussed in Table 2.
3. The provided figures in Table 2 and Table 3 contain multiple inaccuracies in labeling "Best results" and "second-best"， which is confusing.

**Questions:**

1. Can the authors provide qualitative, direct metrics or a feasible evaluation scheme? The core objective of the newly proposed task is text super-resolution guided by solving the text-image hallucination problem. However, the paper also states that no existing metrics can accurately measure the effectiveness in addressing this issue. The proposal of a task with no clear way to determine if it has been completed lacks meaningfulness.
2. Firstly, the comparison involves too few methods, which undermines the credibility of the claimed superiority in image quality. Secondly, it is unclear why most performance metrics for the "further trained DiffBIR" model are significantly lower than those of the original DiffBIR. Moreover, the fact that this model was not included in the comparisons of Tables 2/3。 This counterintuitive result requires clarification.

---

> ### Author Response · Authors · 2025-11-26
>
> Dear Reviewer 5iBr,
>
> We sincerely appreciate your insightful comments and questions, which have greatly contributed to improving our work. We address your concerns as follows.
>
> ---
>
> > **[W1 - (1)] About metrics**
>
> [A1] We appreciate this comment and acknowledge that while we mentioned the evaluation metrics in Sec. 5.1 and Appendix D, the explanation could have been more comprehensive. We clarify that the text-spotting F1-score metrics do assess linguistic correctness, as shown in the End-to-End (None and Full) columns of Tab. 2 and 3.
>
> Specifically, the evaluation operates in two stages: (1) text detection is assessed using Precision, Recall, and F1-score, and (2) for detected text regions, End-to-End metrics evaluate recognition accuracy by comparing transcribed text against ground-truth annotations using exact string matching. This process directly measures linguistic correctness; if restoration produces inaccurate text, End-to-End F1-scores will be lower. This evaluation protocol follows standard practice in text-spotting benchmarks.
>
> In the revision, we will provide a more detailed explanation in Sec. 5.1, making the connection between End-to-End F1-scores and restoration fidelity more explicit.
>
> > **[W1 - (2)] Qualitative results**
>
> [A2] We would like to clarify that qualitative text restoration results are already provided in Figs. 10–13 in the Appendix. Specifically, Fig. 13 presents the Real-Text results, while Figs. 10–12 demonstrate restoration quality on SA-Text Lv1, 2, and 3. We kindly invite the reviewer to refer to these figures for detailed visual comparisons.
>
> > **[W2] Image metrics**
>
> [A3]We thank the reviewer for this suggestion. We would like to clarify the purpose of Tab. 4, which is not to claim superior general image restoration, but rather to demonstrate that incorporating text-focused objectives does not compromise overall image quality, a critical consideration in multi-task learning. Our primary contribution is introducing text-aware image restoration, focusing specifically on text-focused restoration tasks.
>
> We appreciate the suggestion to include additional comparisons with other methods and we will add these results in Appendix to provide a more comprehensive evaluation of general image quality. Note that while general image quality metrics provide useful reference, they do not fully capture our text-aware restoration objectives, which are primarily evaluated through text-spotting metrics in Tab. 2 and 3.
>
> > **[W3] About Table**
>
> [A4] Thank you for pointing this out. We will fix them in the revised version.
>
> > **[Q2] About further trained DiffBIR**
>
> [A5] Thank you for pointing this out. We apologize for the confusion.
> To clarify, we used the "further trained" DiffBIR weights released by the authors (trained with LLaVA captioner) rather than their original published weights. The lower performance in Tab. 4 is likely due to dataset distribution mismatch: the further trained model was optimized for nature-themed image datasets, while Tab. 4 evaluates on the text image dataset.
>
> Additionally, the 'further trained' DiffBIR are already provided in Tab. 9 and 10 in Appendix. However, we acknowledge that this separation may confuse readers. To improve clarity, we will include ‘further trained’ version in Tab. 2 and 3 in the revision.
>
> ---
> Many thanks to Reviewer 5iBr for the valuable feedback. We have carefully addressed each comment and hope our responses resolve your concerns. Your comments greatly improve the clarity and quality of our work, and we appreciate your time and effort. We’re happy to address any further concerns during the discussion period.

---

### Official Review · Reviewer_Amhv · 2025-10-30

**Soundness:** 2
**Presentation:** 3
**Contribution:** 2
**Rating:** 4
**Confidence:** 4

**Summary:**

This paper presents Text-Aware Image Restoration (TAIR), designed to mitigate diffusion models’ tendency to hallucinate text in degraded images. The authors introduce SA-Text, a dataset of 100K images with detailed text annotations, and TeReDiff, a diffusion-based framework that integrates text spotting for joint training and text-conditioned denoising. Experimental results demonstrate improved text fidelity compared with existing approaches on the SA-Text and TextZoom benchmarks. However, the method’s dependence on synthetic degradations, limited validation on real-world data, and insufficient analysis of computational overhead reduce its practical significance.

**Strengths:**

1. The author clearly identifies a critical limitation in diffusion-based image restoration—its inability to accurately recover text regions. To address this, the author introduces Text-Aware Image Restoration (TAIR) as a novel task that jointly optimizes visual quality and text fidelity, offering strong potential for practical applications.

2. The author constructs a dataset of 100K high-resolution images derived from SA-1B, densely annotated with text polygons and transcriptions through a scalable VLM-based annotation pipeline.

**Weaknesses:**

1. The author primarily conducts training and evaluation on synthetic degradations (e.g., Real-ESRGAN). Moreover, while the Real-Text results (Table 3) demonstrate modest improvements (e.g., a +6% F1-score over FaithDiff), the absence of corresponding visual examples limits the clarity of qualitative gains.

2. The author provides no analysis of error propagation across the pipeline. Given that the method relies on accurate text spotting, recognition errors are likely to propagate and compromise restoration performance, but this crucial aspect remains unexplored.

3. Jointly training the text-spotting module with the diffusion backbone is conceptually straightforward. However, the ablation studies (Table 6) indicate that most of the observed improvements stem from the inclusion of the SA-Text dataset and the use of multi-stage training, rather than from the joint optimization itself.

4. The evaluation depends exclusively on text-spotting metrics (e.g., F1-score), which fail to reflect the semantic accuracy of restored text. Such metrics emphasize detection precision and recall but overlook whether the recovered text is contextually or linguistically correct, thereby limiting the reliability of the reported improvements.

5. Although SA-Text is described as a “language-agnostic” annotation pipeline (Section 3), the experiments and validations are conducted exclusively on English text. The absence of cross-lingual evaluations leaves the claimed language generality unverified.

**Questions:**

1. When the text-spotting module fails at early denoising steps (e.g., misreads "HELLO" as "H3LLO"), how does TeReDiff recover?

2. Ablations suggest SA-Text drives performance. If trained on TextOCR, would TeReDiff still outperform DiffBIR?

3. Why not test on non-English text? Can the VLM filtering in SA-Text handle languages with complex scripts?

---

> ### Author Response · Authors · 2025-11-26
>
> Dear Reviewer Amhv,
>
> We sincerely appreciate your insightful comments and questions, which have greatly contributed to improving our work. We address your concerns as follows.
>
> ---
>
> > **[W1] Qualitative results**
>
> [A1] We would like to clarify that qualitative text restoration results are already provided in Figs. 10–13 in the Appendix, which include comparisons with FaithDiff and other diffusion-based restoration models. Specifically, Fig. 13 presents the Real-Text results, while Figs. 10–12 demonstrate restoration quality on SA-Text Lv.1, 2, and 3. We kindly invite the reviewer to refer to these figures for detailed visual comparisons.
>
> > **[W2, Q1] Error propagation and recovery**
>
> [A2] To directly address the concern about error propagation in our pipeline, we conducted a detailed analysis comparing the impact of textual prompt guidance against the null setting.
>
> |   | SA-Text (Lv1) | SA-Text (Lv2) | SA-Text (Lv3) | Real-Text |
> |---|---------------|---------------|---------------|-----------|
> | A | 44            | 33            | 25            | 39        |
> | B | 53            | 72            | 47            | 74        |
>
> - **Case A (Error Propagation)**: Text instances that were correctly restored in the null setting but failed when using prompts from the text-spotting module, indicating error propagation that the reviewer mentioned.
> - **Case B (Performance Gain)**: Text instances that failed in the null setting but were successfully restored when using prompts from the text-spotting module, demonstrating the benefit of accurate text guidance.
>
> We report the number of text box instances for both cases across SA-Text Lv1, 2, 3, and Real-Text datasets in the above table. Case B consistently and significantly outnumbers Case A across all degradation levels and datasets, demonstrating that the positive impact of textual guidance substantially outweighs occasional errors. This is reflected in Tab. 6(a), where our method consistently outperforms the null condition baseline.
>
> Additionally, as illustrated in Fig. 5, the text-spotting module may occasionally misinterpret texts at early denoising steps but can progressively refine its predictions by leveraging partially restored diffusion features throughout the denoising process. While refinement does not always succeed, this self-correction mechanism, combined with the analysis above, confirms the overall effectiveness of textual-prompt guidance.
>
> > **[W3] About ablation studies**
>
> [A3] We thank the reviewer for this observation and provide a detailed breakdown to clarify the contribution of joint optimization. The table below shows the End-to-End (None) performance on SA-Text Lv2:
>
> | Model              | Text condition     | E2E (None) |
> |--------------------|--------------------|------------|
> | DiffBIR^{dagger}   | null               | 16.15      |
> | Stage1             | null               | 21.24      |
> | Stage3             | Captioner (Ours)   | 26.39      |
>
> - **SA-Text dataset contribution** (Row 1 → Row 2): +5.09 improvement comes from training on SA-Text.
> - **Joint optimization contribution** (Row 2 → Row 3): +5.15 improvement comes from jointly training the text-spotting module (TSM) with the diffusion backbone. During Stage 3, the TSM is further refined through joint optimization, producing more accurate textual guidance, while the diffusion model simultaneously learns text-aware features that enhance restoration quality.
>
> This analysis demonstrates that joint optimization contributes as much as the SA-Text dataset to the overall performance gain, and thus cannot be considered a minor factor. Both components are essential to our method's effectiveness.
>
> > **[W4] About metrics**
>
> [A4] We appreciate this comment and acknowledge that while we mentioned the evaluation metrics in Sec. 5.1 and Appendix D, the explanation could have been more comprehensive. We clarify that the text-spotting F1-score metrics do assess linguistic correctness, as shown in the End-to-End (None and Full) columns of Tabs. 2 and 3.
>
> Specifically, the evaluation operates in two stages: (1) text detection is assessed using Precision, Recall, and F1-score, and (2) for detected text regions, End-to-End metrics evaluate recognition accuracy by comparing transcribed text against ground-truth annotations using exact string matching. This process directly measures linguistic correctness, if restoration produces inaccurate text, End-to-End F1-scores will be lower. This evaluation protocol follows standard practice in text-spotting benchmarks.
>
> In the revision, we will provide a more detailed explanation in Sec. 5.1, making the connection between End-to-End F1-scores and restoration fidelity more explicit.

---

> ### Author Response · Authors · 2025-11-26
>
> > **[W5, Q1] About language-agnostic**
>
> [A5] Thank you for raising this important point regarding language generality.
>
> Our SA-Text annotation pipeline is language-agnostic by design, as it leverages VLM's multilingual capabilities for automatic annotation. During development, we observed the VLM successfully handling complex scripts like Chinese, demonstrating the pipeline's potential for multilingual dataset construction.
>
> While our dataset curation pipeline can support multiple languages, we trained the text spotting module (TSM) exclusively on English. This decision was made to maintain focus on our core contribution. As the first work to learn a TSM from diffusion features, we prioritized validating this novel learning paradigm with English to ensure clear, interpretable results, rather than addressing the substantial complexities of multilingual text spotting [1].
>
> Training separate TSMs for multiple languages would require individual models, datasets, and extensive experiments for each language, valuable but orthogonal to our core contribution. Our method is fundamentally language-agnostic and provides a solid foundation for future multilingual extensions.
>
> > **[Q2] TextOCR Dataset**
>
> [A6] We note that TextOCR is fundamentally unsuitable as an image restoration training dataset. As described in Sec.3 and Tab.1, TextOCR was designed for text recognition in natural scenes, containing only 21K low-quality images without the high-quality references.
>
> In contrast, SA-Text100K was specifically curated for text image restoration with 100K high-quality image pairs. While TextOCR could provide supervisory signals for training TeReDiff, its limited scale and low image quality would result in a model substantially inferior to one trained on SA-Text100K, which offers both the scale and quality essential for learning robust text-aware restoration features.
>
> ---
> Many thanks to Reviewer Amhv for the valuable feedback. We have carefully addressed each comment and hope our responses resolve your concerns. Your comments greatly improve the clarity and quality of our work, and we appreciate your time and effort. We’re happy to address any further concerns during the discussion period.
>
> ## **Reference**
> [1] Huang, Jing, et al. "A multiplexed network for end-to-end, multilingual OCR." Proceedings of the IEEE/CVF conference on computer vision and pattern recognition. 2021.

---

> > ### Comment · Reviewer_Amhv · 2025-11-28
> >
> > Thank you for the authors' response.
> >
> > For Weakness 1, I believe the current paper only provides Fig. 13 as evidence for Real-Text results, and the number of examples is very limited, making it insufficient to demonstrate effectiveness. Moreover, some examples contain errors. For instance, in the last row of Fig. 13, "ISC0016" is incorrectly restored as "15C0016". Therefore, how the framework performs on real-world data remains unclear and requires further justification.
> >
> > For Weakness 2, although the number of Case B instances is larger than Case A, the difference is not substantial, and Case A still accounts for a large portion. For example, on SA-Text (Lv1), the ratio is 44/(44+53) = 45.36%, which is nearly half. Thus, the conclusion drawn from this experiment does not appear to be very convincing.
> >
> > For Weakness 3, 4 and Q6, I agree with the authors' explanations.
> >
> > For Weakness 5, since no sufficient experiments were conducted to validate the language-agnostic claim, this point should not be emphasized in the paper.

---

> ### Author Response · Authors · 2025-12-03
>
> We appreciate the reviewer's thoughtful follow-up. We address the additional concerns below.
>
> ---
>
> > **[Comment 1] Real-world results**
>
> We would like to respectfully clarify that our evaluation on real-world data extends beyond Fig. 13. Specifically, we report results on the TextZoom dataset, a widely adopted real-world benchmark in Scene Text Image Super-Resolution (STISR), in Tab. 5 and Fig. 16. As shown in Tab. 5, our method achieves substantial performance gains over TextSR, a concurrent state-of-the-art work:
>
> - CRNN: 58.7% → 72.4% (+13.7pp, +23.3%)
> - MORAN: 64.0% → 85.0% (+21.0pp, +32.8%)
> - ASTER: 65.8% → 82.2% (+16.4pp, +24.9%)
>
> Regarding the specific case noted by the reviewer, while our method may produce minor errors in such challenging samples, baseline methods tend to generate unrecognizable text-like patterns (text-image hallucinations) for the same inputs. The Full Lexicon results in Tab. 3 confirm that our method consistently produces more accurate text. To further demonstrate real-world effectiveness, we will include additional qualitative comparisons in the revised version.
>
> We also wish to clarify the Real-Text quantitative results in Tab. 3. It is unclear which specific metric the reviewer's "+6% F1-score" refers to, so we provide a breakdown of the End-to-End (None) recognition results.
>
> **Compared to FaithDiff (as mentioned by the reviewer):**
>
> - ABCNetv2: 38.81% → 48.39% (+9.58pp, +24.7%)
> - TESTR: 41.64% → 49.39% (+7.75pp, +18.6%)
>
> **Compared to StableSR (second-best):**
>
> - ABCNetv2: 41.23% → 48.39% (+7.16pp, +17.4%)
> - TESTR: 42.53% → 49.39% (+6.86pp, +16.1%)
>
> Notably, all baseline methods produce recognition scores lower than directly evaluating on low-quality (LQ) images due to text-image hallucinations. By explicitly addressing this issue, our method is the only one that surpasses the LQ baseline, demonstrating faithful restoration.
>
> Finally, following standard practice in diffusion-based image restoration, we train on the Real-ESRGAN degradation pipeline and evaluate generalization to real-world benchmarks. Our contribution is not a specialized framework for real-world degradations, but a novel approach leveraging text spotting modules for faithful text restoration. The results on TextZoom and Real-Text validate that this approach generalizes well beyond synthetic training data.
>
> > **[Commnet2] Error Propagation**
>
> |   | SA-Text (Lv1) | SA-Text (Lv2) | SA-Text (Lv3) | Real-Text |
> |---|---------------|---------------|---------------|-----------|
> | A | 37            | 31            | 21           | 62        |
> | B | 56            | 65            | 52            | 87        |
>
> To clarify, our claim is not that error propagation is negligible, but rather that **the overall benefit of textual guidance outweighs the occasional errors it introduces,** a trade-off that consistently favors our approach.
>
> Regarding the relatively higher Case A ratio in SA-Text (Lv1), at the mild degradation level, the text structure remains partially recognizable, allowing the model to leverage the provided prompt for restoration actively. Consequently, when the prompt contains errors, the model can more readily act on that erroneous guidance, amplifying the impact of prompt errors compared to more severely degraded cases. We also observed a consistent trend when using ABCNet v2, as provided in the above table.
>
> Importantly, the net effect is reflected in the quantitative results: Tab. 6(a) and Tab. 11 show consistent performance gains over the null results across all degradation levels, confirming that the benefits of textual guidance outweigh the costs of occasional errors. Additionally, we were aware of this error propagation and explored a mitigation strategy using a VLM for prompt correction in Appendix G, which yielded further improvements.
>
> > **[Comment3] Language-agnostic**
>
> We appreciate this feedback. To demonstrate the pipeline's language-agnostic capability, we have conducted an additional experiment on Chinese and included the results in Appendix A. However, we agree with the reviewer that this alone does not fully validate the claim, and have therefore toned down the statement in the revised manuscript.

---

### Official Review · Reviewer_YnwB · 2025-11-01

**Soundness:** 2
**Presentation:** 3
**Contribution:** 3
**Rating:** 6
**Confidence:** 5

**Summary:**

This paper proposes TeReDiff, a text-aware image restoration framework that effectively recovers readable text in severely degraded full-scene images while preserving overall visual quality. Unlike prior scene text image super-resolution (STISR) methods that focus on small, single-word crops, TeReDiff addresses the more challenging and general Text-Aware Image Restoration (TAIR) task. By incorporating multi-stage training and text-conditioned diffusion, the model mitigates text hallucination and achieves good performance on both text detection/recognition metrics and general image quality benchmarks.

**Strengths:**

- Introduces TAIR—a more realistic and general setting than existing STISR—by restoring full-scene images with multiple, diverse text instances.
- Significantly reduces text hallucination and improves readability through text-conditioned diffusion and multi-stage training.
- Achieves good performance on both text recognition metrics and standard image restoration benchmarks, demonstrating balanced fidelity and perceptual quality.

**Weaknesses:**

- The evaluation relies heavily on synthetic or curated datasets (e.g., SA-Text), which may not fully reflect the complexity and variability of real-world degraded images.
- Lacks in-depth discussion of scenarios where the method fails (e.g., extremely low-resolution or occluded text), reducing insight into limitations.

**Questions:**

Please refer to Weaknesses.

---

> ### Author Response · Authors · 2025-11-26
>
> Dear Reviewer YnwB,
>
> We sincerely appreciate your insightful comments, which have greatly contributed to improving our work. We address your concerns as follows.
>
> ---
> > **[W1] Results for real-world degradations**
>
> [A1] As outlined in Section 5.1 in the main paper,  we already provide the evaluation of our method on two real-world datasets: Real-Text and TextZoom. The Real-Text dataset is constructed from real-world HR–LR image pairs derived from RealSR and DRealSR, whereas TextZoom also comprises real-world cropped image pairs commonly used in scene text image super-resolution (STISR). Quantitative restoration results for these datasets are presented in Tab. 3 and Tab. 5 of the main paper. Complementary qualitative comparisons are provided in Fig. 13 and Fig.14 from Appendix, further highlighting the effectiveness of our approach in restoring text from real-world degraded inputs.
>
> > **[W2] Fail case analysis**
>
> [A2] We appreciate the reviewer highlighting this important limitation discussion. To analyze the mentioned failure cases, we examined the "None" results from Table 2's TESTR evaluation and report the minimum text sizes in this table. Specifically, this table shows the minimum bounding box sizes of text that was either correctly recognized by the text spotting module at the initial denoising step (Case 1) or accurately restored by our model (Case 2).
>
> |       | SA-Text Level1 | SA-Text Level2 | SA-Text Level3 | Real-Text |
> |-------|----------------|----------------|----------------|-----------|
> | Case1 | 16x17          | 16x17          | 28x22          | 20x16     |
> | Case2 | 14x27          | 25x20          | 35x26          | 20x17     |
>
> We statistically observed a tendency for smaller text to be either more difficult to spot (Case 1) or harder to successfully restore (Case 2). However, various types of noise are randomly applied by the degradation kernel from Real-ESRGAN, and other additional factors such as the specific font and text length may also affect the restoration results. The experiments in this table help identify the practical boundaries of text size at which the text spotting module fails to detect text and the restoration module fails to recover it under various degradation scenarios.
>
> ---
>
> Many thanks to Reviewer YnwB for the valuable feedback. We have carefully addressed each comment and hope our responses resolve your concerns. Your comments greatly improve the clarity and quality of our work, and we appreciate your time and effort. We’re happy to address any further concerns during the discussion period.

---

### Author Response · Authors · 2025-11-26
**Paper Revision Summary**

We sincerely thank the Area Chair and all reviewers for their time and effort in reviewing our work, as well as their valuable feedback and constructive comments. We are pleased that reviewers recognized the originality and practical significance of TAIR task (YnwB, Amhv, 5iBr), the meaningful contribution of our large-scale SA-Text dataset to the community (Amhv, 5iBr, cLfC), and the strong performance of our method across multiple benchmarks (YnwB, 5iBr, cLfC). We also appreciate the positive remark on the paper's clarity (5iBr).

In response to the reviewers' concerns, we have conducted all necessary clarifications and additional experiments. These have been incorporated into the revised manuscript, with the key updates summarized as follows:

- **Overall Layout**: We improved the layout to enhance readability by placing referenced content (tables and figures) closer to the corresponding text.
- **Sec. 1**: We acknowledge the concurrent work TADiSR and have revised our discussion to clarify the key differences between our approach and TADiSR.
- **Sec. 3 and Appendix A**: We added the results of our pipeline on Chinese in Appendix and toned down the language-agnostic claim in the main text.
- **Sec. 5.1**: We added a detailed explanation of how End-to-End metrics operate and measure recognition accuracy through linguistic correctness evaluation.
- **Sec. 5.2 and Appendix E**: Following the reviewers' feedback, we provide additional context on image quality metrics to clarify that our primary focus is text restoration while maintaining competitive overall image quality. We also include comprehensive image quality comparisons with various methods in Appendix E.
- **Appendix F**: We provided additional qualitative comparisons on Real-Text and TextZoom to validate effectiveness in real-world scenarios.
- **Appendix G**: We provide an analysis of failure cases by examining the minimum bounding box sizes that the text-spotting module can successfully recognize and that TeReDiff can effectively restore.

---

We believe these revisions address the reviewers' concerns and improve the overall quality of our manuscript. We sincerely thank the reviewers for their thoughtful feedback and valuable suggestions.

---

### Author Response · Authors · 2025-12-03
**Key Contributions & Discussion Summary**

We sincerely thank the ACs and reviewers for their thorough review and constructive feedback, which have greatly helped improve our manuscript. Below, we provide a summary of our contributions and how we addressed the reviewers' concerns.

---
## **Key Contributions**

- We propose **TAIR (Text-Aware Image Restoration)**, a new task for restoring images while faithfully recovering textual regions, addressing the text-image hallucination problem in existing diffusion-based restoration methods.
- We also present **SA-Text, a large-scale benchmark of 100K high-quality scene images** with dense text annotations.
- We introduce **TeReDiff, a multi-task diffusion framework that jointly trains a text-spotting module (TSM)** with the restoration module. By leveraging internal diffusion features for text spotting, intermediate text predictions can condition the restoration process during denoising, enabling faithful text recovery.

---
## **Discussion Summary**
We have addressed the reviewers' concerns with text revisions and additional experiments, summarized as follows:

- **Text-spotting metrics**:
  - Reviewer Amhv, 5iBr asked whether text-spotting metrics can measure the linguistic correctness of restored text.
  - We clarified that End-to-End F1-score metrics assess linguistic correctness through exact string matching, and added a more detailed explanation in Sec. 5.1. This was acknowledged by Reviewer Amhv.
- **Real-world degradation results**
  - Reviewer YnwB, cLfc, 5iBr requested results on real-world degradation.
  - We first clarified that results on real-world degradation datasets, Real-Text and TextZoom, were already provided in the Appendix. To further validate our framework, we also added more qualitative results on these datasets.
- **Language-agnostic**
  - Reviewer Amhv, cLfc asked about the applicability to other languages.
  - We provided a detailed explanation of why our dataset pipeline and framework are language-agnostic, and why we focus on English in this paper. Additionally, we included the results of our pipeline on Chinese in the Appendix.
- **Joint optimization**
  - Reviewer Amhv asked about the effectiveness of joint optimization.
  - We provided a detailed breakdown to clarify its contribution. This was acknowledged by Reviewer Amhv.

---

We believe our responses and revised manuscript address the reviewers' concerns. We again thank the ACs and reviewers for their time and dedication.

---

### Meta-Review · Area_Chair_SAYL · 2025-12-15

**Summary:**

The paper proposes text-aware image restoration to address the tendency of diffusion models to hallucinate text patterns. The authors introduce SA-Text, a large-scale benchmark of 100K scene images with dense text annotations , and TeReDiff, a framework that jointly trains a text-spotting module to guide the restoration process via OCR-based prompts. While prior work like TextSR focuses on Scene Text Image Super-Resolution for cropped inputs , this paper generalizes the problem to full-scene images containing complex backgrounds and multiple text instances. The method demonstrates strong performance on the TextZoom benchmark (a subtask of TAIR).

**Reviewer Concerns:**

**Addressed Reviewer Concerns:**
 - Evaluation Metrics: The authors successfully clarified that End-to-End F1-score metrics measure linguistic correctness through exact string matching, a point acknowledged by Reviewers Amhv and 5iBr.

- Dataset Availability: The authors confirmed the release of the code, weights, and the SA-Text dataset, securing the contribution value.

- Concurrent Work: The authors distinguished their work from concurrent full-scene methods (e.g., TADISR) by highlighting their use of linguistic OCR outputs versus segmentation maps.

**Outstanding Concerns:**
- Real-World Generalization: Despite providing additional qualitative results, Reviewer Amhv identified specific hallucinations in the rebuttal examples (e.g., "ISC0016" restored as "15c0016"). This suggests the model, trained on synthetic Real-ESRGAN data, still struggles with real-world artifacts.

- Error Propagation: Reviewer Amhv noted that in mild degradation scenarios (SA-Text Lv1), error propagation from the text-spotting module remains significant (approx. 45%), meaning erroneous prompts can actively hamper restoration.

- Language-Agnostic Claims: The claim that the method is "language-agnostic" was unsupported by experiments (as the model was trained only on English), which the authors conceded to tone down.

**Reviewer Scores:**

- **Reviewer YnwB** (Initial Score: 6): Likely maintains positive score. They found the fail case analysis regarding text size limitations sufficient.

- **Reviewer Amhv** (Initial Score: 4): Likely maintains negative score. They remained unconvinced by the real-world qualitative results and the error propagation trade-off.

- **Reviewer 5iBr** (Initial Score: 6): Likely maintains positive score. Their concerns about metrics were addressed, and they valued the dataset originality.

- **Reviewer cLfC** (Initial Score: 4): Likely maintains negative score. They viewed the architectural novelty (integrating ControlNet) as limited and the real-world evaluation as insufficient.

---

### Decision · Program_Chairs · 2026-01-26

Accept (Poster)